# The Alzheimer's disease-associated C99 fragment of APP regulates cellular cholesterol trafficking

Jorge Montesinos[1,†], Marta Pera[1,†,‡], Delfina Larrea[1,†], Cristina Guardia-Laguarta[1], Rishi R Agrawal[2], Kevin R Velasco[1], Taekyung D Yun[1], Irina G Stavrovskaya[1], Yimeng Xu[3], So Yeon Koo[4], Amanda M Snead[4], Andrew A Sproul[4,5] & Estela Area-Gomez[1,2,4,*]

## Abstract

The link between cholesterol homeostasis and cleavage of the amyloid precursor protein (APP), and how this relationship relates to Alzheimer's disease (AD) pathogenesis, is still unknown. Cellular cholesterol levels are regulated through crosstalk between the plasma membrane (PM), where most cellular cholesterol resides, and the endoplasmic reticulum (ER), where the protein machinery that regulates cholesterol levels resides. The intracellular transport of cholesterol from the PM to the ER is believed to be activated by a lipid-sensing peptide(s) in the ER that can cluster PM-derived cholesterol into transient detergent-resistant membrane domains (DRMs) within the ER, also called the *ER regulatory pool of cholesterol*. When formed, these cholesterol-rich domains in the ER maintain cellular homeostasis by inducing cholesterol esterification as a mechanism of detoxification while attenuating its *de novo* synthesis. In this manuscript, we propose that the 99-aa C-terminal fragment of APP (C99), when delivered to the ER for cleavage by γ-secretase, acts as a lipid-sensing peptide that forms regulatory DRMs in the ER, called mitochondria-associated ER membranes (MAM). Our data in cellular AD models indicates that increased levels of uncleaved C99 in the ER, an early phenotype of the disease, upregulates the formation of these transient DRMs by inducing the internalization of extracellular cholesterol and its trafficking from the PM to the ER. These results suggest a novel role for C99 as a mediator of cholesterol disturbances in AD, potentially explaining early hallmarks of the disease.

**Keywords** Alzheimer's disease; amyloid precursor protein; cholesterol; lipid rafts; mitochondria-associated ER membranes
**Subject Categories** Membrane & Trafficking; Neuroscience
**The EMBO Journal (2020) 39: e103791**

## Introduction

The lipid composition of cellular membranes undergoes continuous modulation to regulate processes such as signal transduction and transmembrane ion gradients (Lauwers *et al*, 2016). To support these events, a network of enzymes interconnects the metabolism of all lipids and promotes the remodeling of membranes into functional subregions (Lingwood & Simons, 2010). Often, these domains display the characteristics of lipid rafts, or detergent-resistant domains (Lingwood & Simons, 2010). Lipid rafts are transient membrane subregions formed by local increases in free or unesterified cholesterol, shielded from the aqueous phase through interaction with sphingomyelin (SM) and saturated phospholipids (Lingwood & Simons, 2010). These local elevations in cholesterol create highly ordered membrane microdomains that passively segregate and enrich for lipid-binding proteins, facilitating protein–protein interaction(s) and regulation of specific signaling pathways (Lingwood & Simons, 2010). Formation of these domains is enabled by "lipid-sensing" proteins with the capacity to bind and cluster cholesterol (Epand *et al*, 2006). If these lipid-sensing proteins were to be eliminated, local cholesterol clusters would disperse, dissolving lipid rafts into the liquid-disordered state. A critical event in this process is the activation of sphingomyelinases (SMases), which hydrolyze SM to ceramide (Chang *et al*, 2006). As opposed to SM, ceramide creates an electrostatically unfavorable environment for cholesterol (Yu *et al*, 2005) and makes cholesterol accessible for removal from the membrane via, for example, esterification (Chang *et al*, 2006). Hence, the turnover of lipid raft domains and their capacity to regulate signaling pathways are tightly linked to the regulation of cholesterol homeostasis. Thus, alterations in cholesterol metabolism would affect lipid raft formation, and vice versa.

Cellular cholesterol is either synthesized *de novo* in the endoplasmic reticulum (ER) or taken up as cholesteryl esters (CEs) from lipoproteins (Chang *et al*, 2006). The *de novo* synthesis of

1   Department of Neurology, Columbia University Irving Medical Center, New York, NY, USA
2   Institute of Human Nutrition, Columbia University Irving Medical Center, New York, NY, USA
3   Biomarkers Core Laboratory, Columbia University Irving Medical Center, New York, NY, USA
4   Taub Institute for Research on Alzheimer's Disease and the Aging Brain, Columbia University Irving Medical Center, New York, NY, USA
5   Department of Pathology and Cell Biology, Columbia University Irving Medical Center, New York, NY, USA
    *Corresponding author. Tel: +1 212 305 1009; E-mail: eag2118@columbia.edu
    †These authors contributed equally to this work
    ‡Present address: Basic Sciences Department, Faculty of Medicine and Health Sciences, Universitat Internacional de Catalunya, Barcelona, Spain

cholesterol is activated by the transport of the sterol regulatory element binding protein isoform 2 (SREBP2, gene *SREBF2*) from the ER to the Golgi and its subsequent activation by proteolytic cleavage (Brown & Goldstein, 1999). The processed form of SREBP2 translocates to the nucleus and induces the transcription of cholesterol-synthesizing genes as well as that of *SREBF2* itself (Brown & Goldstein, 1999). When cellular cholesterol supply is sufficient, SREBP2 is retained in the ER in its uncleaved form, preventing activation of the *de novo* cholesterol synthesis pathway (Brown & Goldstein, 1999). When taken up from extracellular lipoproteins, internalized CEs are hydrolyzed in endolysosomes to unesterified free cholesterol, most of which is transferred to the plasma membrane (PM) (Das *et al*, 2014; Infante & Radhakrishnan, 2017). Once the PM cholesterol concentration surpasses a threshold, it is transported to the ER for esterification by the enzyme, acyl-coenzyme A:cholesterol acyltransferase 1 (ACAT1; gene *SOAT1*), as a means of detoxifying the excess cholesterol (Das *et al*, 2014). The resultant CEs are stored in lipid droplets in the cytosol before being secreted (Das *et al*, 2014). Thus, to preserve cholesterol homeostasis, the cell maintains crosstalk between the PM, where the bulk of the cell's cholesterol resides, and the ER, where the enzymatic activities that regulate cholesterol levels reside (Litvinov *et al*, 2018). This crosstalk is believed to be controlled by an as-yet-unknown sensor in the ER that triggers communication between the PM and the intracellular ER regulatory pool of cholesterol where ACAT1 is located (Lange *et al*, 1999; Infante & Radhakrishnan, 2017).

The regulation of lipid metabolism, including that of cholesterol, is particularly critical within the nervous system (Petrov *et al*, 2016). It is therefore not surprising that lipid dysregulation has been described in multiple neurodegenerative diseases, including Alzheimer's disease (AD) (Di Paolo & Kim, 2011). Specifically, cholesterol abnormalities in AD have been widely reported (Di Paolo & Kim, 2011), but the field currently lacks consensus as to their cause(s).

The "amyloid cascade hypothesis" of AD pathogenesis states that increases in the levels of the β-amyloid peptide (Aβ), derived from APP processing, trigger neurodegeneration (Goedert & Spillantini, 2006). In addition to these higher levels of Aβ, AD samples also present with increased cleavage of endocytosed full-length APP by β-secretase (BACE1) to produce the immediate precursor of Aβ, the 99-aa C-terminal domain of APP (C99) (Goedert & Spillantini, 2006). These alterations in APP metabolism are due to mutations in the *PSEN1* [presenilin-1 (PS1)], *PSEN2* [presenilin-2 (PS2)], and *APP* genes in familial AD (FAD), or by unknown causes in sporadic cases (SAD) (Goedert & Spillantini, 2006). Further linking AD and cholesterol, reciprocal modulation between cellular APP distribution and membrane cholesterol concentration has been recently reported (DelBove *et al*, 2019). Moreover, processing of APP C-terminal fragments (APP-CTFs) occurs in detergent-resistant membranes (DRMs) (Cordy *et al*, 2006). Notably, elevations in C99 have been shown to contribute to AD pathology (Shen & Kelleher, 2007), causing endosomal dysfunction (Jiang *et al*, 2010) and hippocampal degeneration (Lauritzen *et al*, 2012; Pulina *et al*, 2020).

Previously, we and others found that C99, when delivered to the ER for cleavage by γ-secretase, is not distributed in the ER homogeneously but is concentrated in mitochondria-associated ER membranes (MAMs) (Area-Gomez *et al*, 2009; Newman *et al*, 2014;

Schreiner *et al*, 2015; Del Prete *et al*, 2017; Pera *et al*, 2017). MAM is a DRM/lipid raft subdomain within the ER (Hayashi & Fujimoto, 2010; Area-Gomez *et al*, 2012) involved in the regulation of lipid homeostasis (Vance, 2014). We showed that AD cell and animal models display an increase of C99 at MAM (Pera *et al*, 2017) that results in the upregulation of MAM activities (Area-Gomez *et al*, 2012; Hedskog *et al*, 2013), including SMases and cholesterol esterification by ACAT1 (Pera *et al*, 2017), a known MAM-localized enzyme (Chang *et al*, 2009). Remarkably, inhibition of C99 production caused the inactivation of these MAM functions (Pera *et al*, 2017).

We now report that in AD, by means of C99's affinity for cholesterol (Barrett *et al*, 2012), the pathogenic accumulation of C99 in the ER (Pera *et al*, 2017) induces the uptake of above-normal levels of extracellular cholesterol. Trafficking of this excess cholesterol from the PM to the ER results in the continuous formation, activation, and turnover of MAM domains, previously observed in cells from AD patients (Area-Gomez *et al*, 2012; Pera *et al*, 2017). Altogether, our data suggest a pathogenic role for C99 elevations in AD, via upregulation of cholesterol trafficking and MAM activity, which disrupt cellular lipid homeostasis and cause the alterations in membrane lipid composition commonly observed during AD pathogenesis.

# Results

## Accumulation of C99 in the ER triggers cellular cholesterol uptake and trafficking to MAM

Recently, we found that increases in the levels of uncleaved C99 at MAM cause the co-activation of SMase(s) and cholesterol esterification via ACAT1 (Pera *et al*, 2017), a mechanism by which cells "detoxify" membranes from an excess of cholesterol (Lange *et al*, 1999). In light of these data, we hypothesized that the upregulation of sphingolipid turnover and cholesterol esterification in cell models of AD could be caused by elevated membrane cholesterol levels, which would activate this detoxification pathway.

To test this, we measured the concentration of cholesterol in homogenates and subcellular fractions from mouse embryonic fibroblasts (MEFs) null for both *PSEN1* and *PSEN2* (PS-DKO) (Herreman *et al*, 2000) [these cell display high levels of C99 in MAM (Pera *et al*, 2017)] (Fig 1A) and from homogenates of AD fibroblasts (Fig EV1A) by liquid chromatography–mass spectrometry (LC-MS) (Chan *et al*, 2012). We found that these models displayed increased levels of free cholesterol compared to controls. This increase in membrane-bound cholesterol was highly significant in total homogenates and MAM membranes (Fig 1A) which, in light of previous data (Area-Gomez *et al*, 2012), could indicate that the upregulation of cholesterol esterification in AD cell models is the result of cholesterol buildup in membranes and its subsequent elimination by esterification. We were able to recapitulate this result in MEFs in which both *APP* and its paralog *APLP2* were knocked out (APP-DKO) (Zhang *et al*, 2013), transiently transfected with a plasmid expressing C99 (Fig 1B). Conversely, APP-DKO cells expressing either the APP-C83 peptide (produced by the cleavage of APP by α-secretase), the AICD peptide (produced by cleavage of C99 by γ-secretase), or incubated with amyloid Aβ$_{42}$ oligomers did not show these cholesterol elevations, suggesting

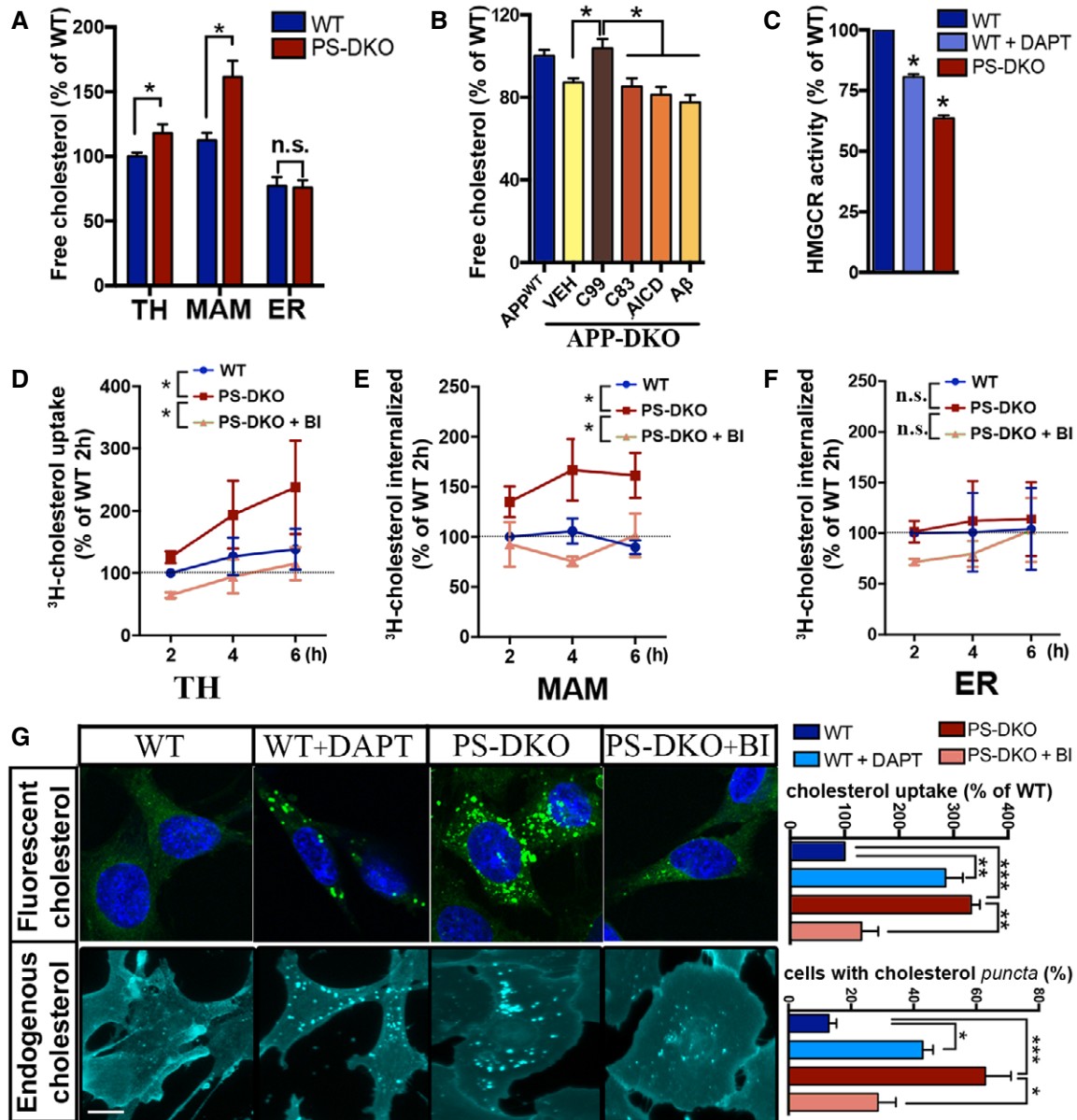

**Figure 1. C99 accumulation promotes cholesterol uptake and trafficking to MAM.**

A    Quantification of free cholesterol levels by lipidomics analysis of total homogenates (TH), ER, or MAM fractions isolated from WT and PS-DKO cells. Lipid units are represented as molar mass over total moles of lipids analyzed (mol %). Graphs represent fold change over controls. Unpaired *t*-test ($n = 6$ for TH, $n = 3$ for MAM or ER; *$P < 0.05$).

B    Quantification of free cholesterol levels in total homogenates of APP-DKO cells transiently expressing C99, C83, or AICD peptides, or treated with 5 μM Aβ$_{42}$ oligomers for 16 h. Cholesterol levels of WT cells are shown as a control. Lipid units are represented as molar mass over total moles of lipids analyzed (mol %). Graphs represent fold change over controls. One-way ANOVA ($n = 4$; *$P < 0.05$).

C    Quantification of HMGCR enzymatic activity in PS-DKO and DAPT-treated WT cells. Unpaired *t*-test ($n = 3$; *$P < 0.05$).

D    Quantification of cholesterol uptake by incubating WT and PS-DKO cells, treated or untreated with BACE inhibitor IV (BI, 100 nM), with 2.5 μCi/ml $^3$H-cholesterol for the indicated times. Graph represents cholesterol radioactivity levels in total cell homogenates (TH). Two-way repeated measures ANOVA (Time, Group) ($n = 3$ independent experiments; *$P < 0.05$).

E, F    Quantification of $^3$H-cholesterol delivery to MAM (E) or bulk ER (F) after the indicated times. Control and BI-treated or untreated PS-DKO cells incubated with 2.5 μCi/ml $^3$H-cholesterol for the indicated times, before subcellular fractionation (Western blot of isolated MAM fractions shown in Fig EV1F). Cholesterol delivery to MAM calculated by quantification of radioactivity levels in the isolated fractions. Two-way repeated measures ANOVA (Time, Group) ($n = 3$ independent experiments; *$P < 0.05$).

G    Measurement of cholesterol uptake in the indicated cells by internalization of a fluorescent cholesterol analog [NBD-cholesterol (2 μM; upper panel, nuclei in blue)]. Endogenous free cholesterol was stained with filipin (lower panel). Graphs on the right represent fluorescence intensity (using ImageJ) and the percentage of cells with filipin punctae. Scale bar = 20 μm. One-way ANOVA (30–50 cells/condition from at least 3 independent experiments; *$P < 0.05$, **$P < 0.01$, ***$P < 0.001$).

Source data are available online for this figure.

that C99 affects cholesterol homeostasis but other APP fragments do not (Fig 1B).

Increases in free cholesterol levels could be a result of upregulated *de novo* synthesis, upregulated uptake, or decreased removal. To determine whether cholesterol increases in AD cells occurred via upregulation of the *de novo* cholesterol synthesis pathway, we quantified the activity of the 3-hydroxy-3-methylglutaryl-CoA reductase (HMGCR), the rate-limiting enzyme in the synthesis of cholesterol, in γ-secretase-deficient cells and controls. HMGCR activity was reduced significantly in PS-DKO MEFs, in WT cells treated with γ-secretase inhibitors (DAPT) (Fig 1C), and in AD fibroblasts (Fig EV1B), in agreement with previous observations (Pierrot *et al*, 2013). Consistent with this, the levels of both uncleaved and mature SREBP2 protein were significantly reduced in cholesterol PS-DKO cells and in WT cells treated with DAPT, compared to WT (Fig EV1C).

Ruling out upregulated *de novo* synthesis as a cause of increased cellular cholesterol, we measured the rate of cholesterol uptake in PS-DKO cells and controls through pulse-chase analysis of $^3$H-cholesterol internalization. PS-DKO cells showed an enhanced rate of cholesterol uptake compared to controls (Fig 1D), also observed in AD fibroblasts (Fig EV1D) and in neuronal cells silenced for PS1 alone or for both PS1 + PS2 (Fig EV1E). Interestingly, the increases in cholesterol uptake were abrogated upon BACE1 inhibition (BI), suggesting a role for C99 in the regulation of cholesterol internalization.

As mentioned before, internalized cholesterol is transported to MAM domains in the ER for esterification by ACAT1. To confirm that the internalized cholesterol trafficked to MAM domains in our cell models, we tracked the uptake of $^3$H-cholesterol and its delivery to MAM and/or bulk ER by pulse-chase analysis and subcellular fractionation in PS-DKO cells (Fig EV1F). We found that, in PS-DKO cells, the *rate* of cholesterol incorporation into MAM (Fig 1E) was higher compared to controls or in bulk ER fractions (Fig 1F) and was abrogated upon BACE1 inhibition, confirming an elevated rate of cholesterol influx and mobilization toward the MAM in cells with elevated C99. This enhanced uptake and internalization of extracellular cholesterol was also reflected in the elevated ratio of cholesteryl esters:free cholesterol (CE:FC) (Slotte & Bierman, 1988; Infante & Radhakrishnan, 2017) in PS-DKO cells (Fig EV1G) and in cells from AD patients (Fig EV1H).

To assess whether reductions in the removal of cellular cholesterol in our cell models of AD also contributed to these alterations in cholesterol dynamics, we measured the levels of radiolabeled cholesterol secreted from the cell after the indicated post-incubation times (Fig EV1I). Our data indicate that cholesterol efflux correlates with increased cholesterol uptake, suggesting that, rather than reduced removal, these phenotypes are caused by an increase in cholesterol internalization, esterification, and subsequent efflux from the cell, triggered by increases in C99. Supporting this result, the uptake of fluorescently labeled cholesterol was also increased in both DAPT-treated WT or PS-DKO cells compared to controls (Fig 1G, upper panel) and was abrogated upon BACE1 inhibition, suggesting a role for C99 in the regulation of cholesterol internalization.

To confirm that these results were not an artifact of exposure to exogenous cholesterol and indeed represent upregulated activity of cellular uptake machinery, we stained WT and PS-DKO MEFs cells with filipin, a fluorescent dye that specifically binds to free cholesterol. The *distribution* of free cholesterol, compared to controls, was markedly different in both DAPT-treated WT and PS-DKO MEFs (Fig 1G,

lower panel), as well as in AD fibroblasts vs. controls (Fig EV1J), showing a higher degree of cholesterol *puncta* in the cytosol. Our images also showed that these filipin-positive bodies co-localized with markers of late endosomes (Rab7) and lysosomes (LAMP1), suggesting that these are indeed cholesterol-rich endolysosomes (Fig EV1K).

To extend our observations to neuronal cell models of AD, we measured cholesterol uptake and esterification in WT cultured cortical neurons treated with DAPT, as well as in cultured cortical neurons from a knock-in mouse model carrying a familial mutation in PS1 (PS1$^{M146V}$-KI mice) (Guo *et al*, 1999). As reported previously (Pera *et al*, 2017), cortical neurons from PS1$^{M146V}$-KI mice showed increased C99 levels (Fig EV2A and B) and higher Aβ$_{42}$:Aβ$_{40}$ ratios (Fig EV2C), as well as significant upregulation of MAM activity (Fig EV2D). Similar to PS-DKO cells, neurons from PS1$^{M146V}$-KI mice showed higher uptake of exogenous fluorescently and radiolabeled cholesterol (Fig EV2E upper panel, and F), an increased number of filipin-positive bodies and lipid droplets stained by LipidTox in the cytoplasm (Fig EV2E, middle and lower panels), and a higher level of cholesterol esterification (Fig EV2G). Confirming a role for C99 in the regulation of cholesterol metabolism, these alterations were reversed by inhibition of BACE1 (Fig EV2E).

Finally, we were able to recapitulate these alterations in cholesterol metabolism in induced pluripotent stem cells (iPSCs) in which a pathogenic mutation in APP (London mutation; APP$^{V717I}$) was knocked into both alleles using CRISPR/Cas9. In agreement with our hypothesis, this cell model presented with significant increases in C99 levels (Fig EV2H), upregulated cholesterol uptake and esterification (Fig EV2I and J), and an elevated ratio of cholesteryl esters: free cholesterol (CE:FC) compared to isogenic controls (Fig EV2K).

Altogether, our data indicate that cell models of AD present with increased intracellular cholesterol turnover, triggered by elevations in C99.

## Upregulation of cholesterol uptake induced by elevated C99 results in the activation of sphingomyelinase activity

In cellular membranes, a "regulatory" pool of cholesterol is complexed with SM to shield cholesterol from water and prevent its mobilization (Das *et al*, 2014; Endapally *et al*, 2019). Over a certain threshold of cholesterol concentration, SMases become activated and hydrolyze SM to produce ceramide, releasing the membrane-bound cholesterol for trafficking to the ER (Slotte & Bierman, 1988). Our previous data revealed that increases in MAM-localized C99 trigger the upregulation of SMase(s) activity (Pera *et al*, 2017) and subsequent cholesterol mobilization from the PM to the ER. Therefore, we asked whether the increase in cholesterol mobilization observed in our AD cell models was perhaps a consequence of sustained SMase activity provoked by increases in MAM-C99. To test this idea, we first incubated DAPT-treated WT cells and PS-DKO MEFs, both of which have an accumulation of C99 at MAM (Pera *et al*, 2017), with SMase inhibitors and analyzed cholesterol distribution and esterification by staining with filipin and LipidTox Green, respectively. Interestingly, incubation with SMase(s) inhibitors resulted in a significant reduction in both cholesterol esterification by ACAT1, as evidenced by the amount of lipid droplets (LDs), in PS-DKO cells (Fig 2A), in DAPT-treated SH-SY5Y cells (Fig EV3A–C), and in fibroblasts from AD patients (Fig EV3D). Therefore, we concluded that the increases in ACAT1 activity and

LD production associated with elevated C99 were facilitated by activation of SMase(s), consistent with previous findings (Slotte & Bierman, 1988). However, inhibition of SMase activity was not capable of reducing the internalization of extracellular cholesterol in γ-secretase–deficient cells (Fig 2B and C), which presented with increased numbers of cytoplasmic filipin-positive *punctae*. This result suggests that SMase upregulation in AD models is likely a consequence, rather than the cause, of increased uptake of cholesterol and enrichment of cholesterol in membranes.

We were able to recapitulate this phenotype in DAPT-treated WT mouse cortical neurons treated with SMase inhibitors, which, as before, resulted in reduced lipid droplets (LDs) but failed to rescue the increased uptake of cholesterol, as measured by incubation with fluorescent cholesterol analogs (NBD-cholesterol) and filipin staining (Fig EV3E). Remarkably, treatment with BI was able to revert the sustained uptake and internalization of extracellular cholesterol in our AD models (Fig EV3E), again supporting a role for C99 in this pathway.

Taken together, our results suggest that C99 accumulation in γ-secretase-deficient cells leads to a sustained upregulation of extracellular cholesterol uptake, internalization, and delivery to ER-MAM, where its esterification by ACAT1 is facilitated by SMase(s)-mediated SM hydrolysis (Pera *et al*, 2017).

## Localization of C99 to MAM is dependent on its cholesterol-binding domain

The C99 fragment of APP contains a cholesterol-binding motif within its transmembrane domain (Barrett *et al*, 2012) that could promote its localization to lipid rafts (Beel *et al*, 2010), such as MAM (Area-Gomez *et al*, 2012; Pera *et al*, 2017). To test whether C99's cholesterol-binding domain (CBD) is necessary for its localization to MAM, we transfected APP-DKO cells with plasmids expressing WT C99 and a mutant C99 construct with reduced affinity for cholesterol ($G_{700}AII_{703}G_{704}$ was mutated to $A_{700}AIA_{703}A_{704}$; denoted as $C99^{MUT}$ for simplicity) and treated them with DAPT to impede C99 cleavage by endogenous γ-secretase. After this, we analyzed the subcellular localization of WT and mutant forms of C99 by running membrane homogenates from these same transfected cells through a continuous sucrose density gradient and compared their comigration with compartment-specific markers by Western blot (Fig 3A). As shown previously (Pera *et al*, 2017), $C99^{WT}$ comigrated with endosomal (Rab5) and MAM (Acsl4) markers (Fig 3B), while $C99^{MUT}$ showed markedly reduced comigration with MAM markers (Fig 3C). As expected, flotillin, another protein with affinity for cholesterol, also comigrated with MAM markers in cells transfected with $C99^{WT}$ (Fig 3D). On the other hand, flotillin

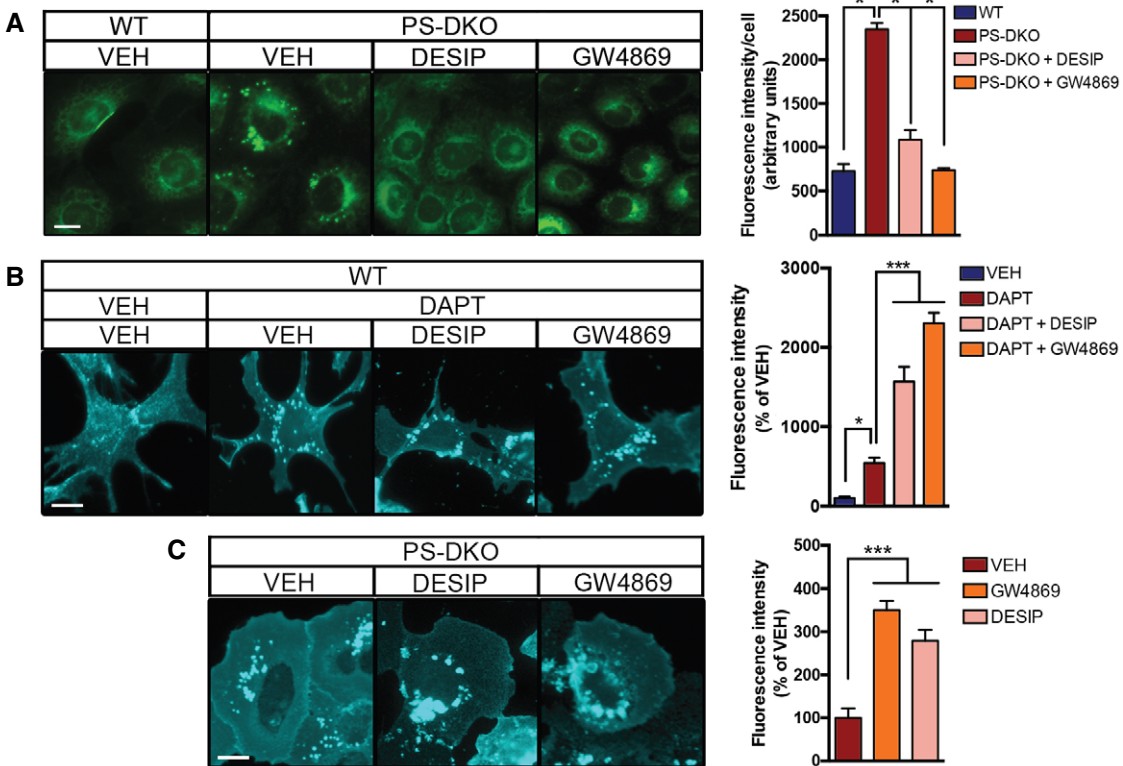

**Figure 2. SMase inhibition prevents lipid droplet generation but fails to rescue the increase in cholesterol uptake caused by C99 accumulation.**

A–C   Representative confocal images of WT and PS-DKO cells treated with the indicated SMase inhibitors (10 μM desipramine or 5 μM GW4869) or DMSO (VEH) for 12–16 h. (A) Endogenous levels of lipid droplets (LDs) were visualized by incubation with LipidTox. Graph shows the LipidTox fluorescence intensity/cell. Endogenous levels of free cholesterol revealed by filipin staining of (B) WT or (C) PS-DKO cells under the indicated treatments. Graphs show the filipin fluorescence intensity as a percentage over VEH. One-way ANOVA (30–50 cells/condition from at least 3 independent experiments; *P < 0.05).

Source data are available online for this figure.

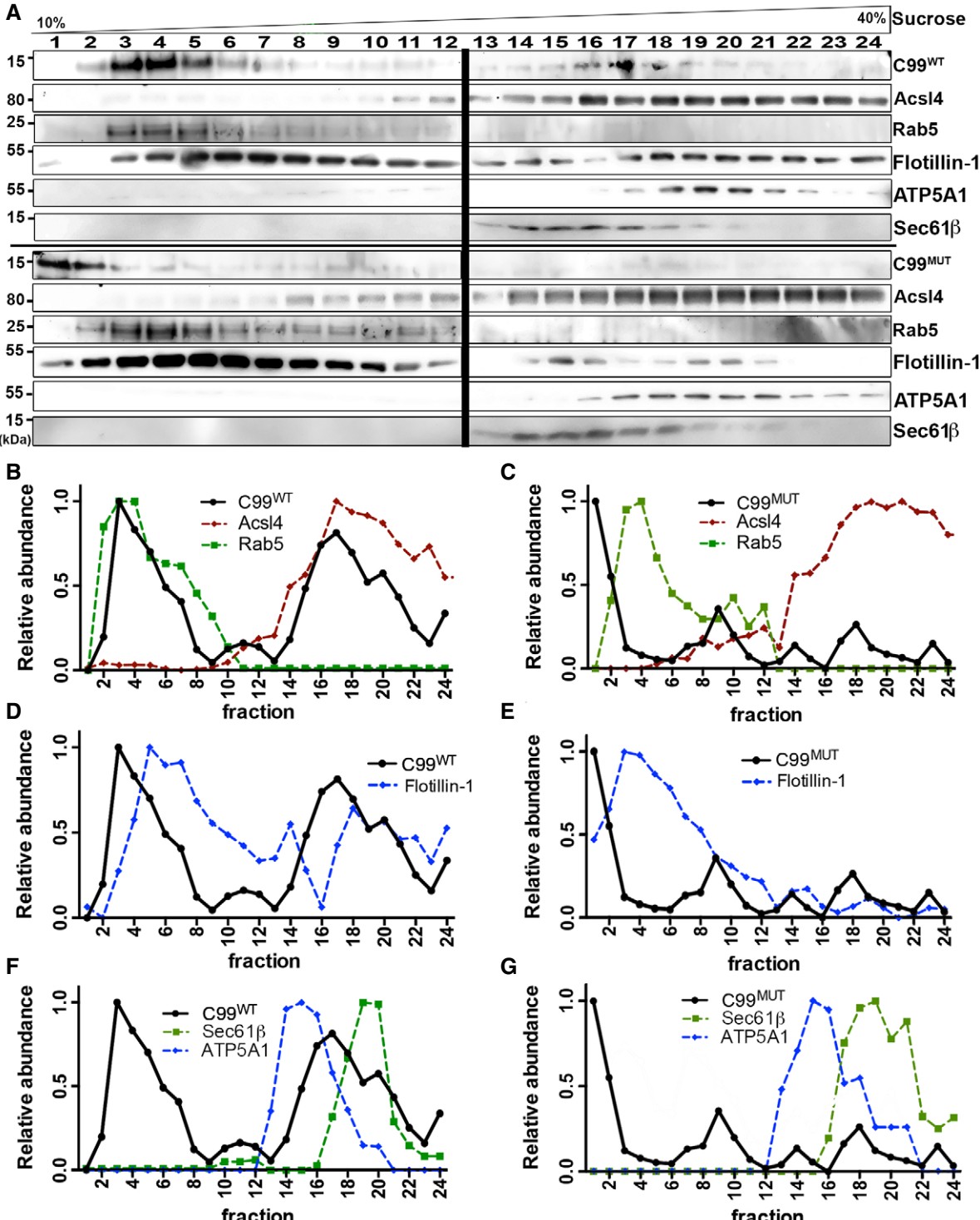

**Figure 3. The cholesterol-binding domain of C99 is necessary for its localization to MAM.**

A Crude membrane fractions from APP-DKO cells expressing C99^WT or C99^MUT were treated with 0.2% Triton X-100, loaded onto continuous density sucrose gradients and centrifuged for 16 h. Fractions from these gradients were analyzed by Western blot to determine the migration of the indicated proteins [two parallel gels (bold vertical line)].

B–G Graphs represent the relative abundance of the indicated proteins in each fraction of the gradient, as measured by densitometry analysis of Western blot signals (ImageJ). Note the reduced degree of comigration of C99^MUT with MAM markers (Acsl4), compared to C99^WT.

Source data are available online for this figure.

showed a reduced comigration with MAM markers in C99[MUT]-expressing cells (Fig 3E). In addition, and contrary to its WT counterpart, C99[MUT] also showed reduced comigration with ER (Sec61β) and mitochondria (ATP5A1) markers (Fig 3F and G), suggesting that the C99 CBD is required for its proper localization to MAM domains.

To substantiate these results, we undertook a confocal imaging approach by transfecting APP-DKO cells with GFP-tagged C99 constructs, alongside fluorescent mitochondrial (MitoDsRed) and ER (Sec 61β-BFP) markers (Fig 4A). While C99[WT] showed a perinuclear

pattern of colocalization with ER and mitochondria, C99[MUT] presented a less marked perinuclear localization and decreased association with mitochondria (Fig 4A and B). Moreover, and as shown before (Pera *et al*, 2017), while overexpression of C99[WT] resulted in significant increases in the apposition between ER and mitochondria (Fig 4C and D), APP-DKO cells expressing C99[MUT] showed no changes in ER–mitochondria connections (Fig 4C and D), and perhaps in MAM formation and activation. This suggests that cells expressing C99[MUT] are deficient for MAM functionality.

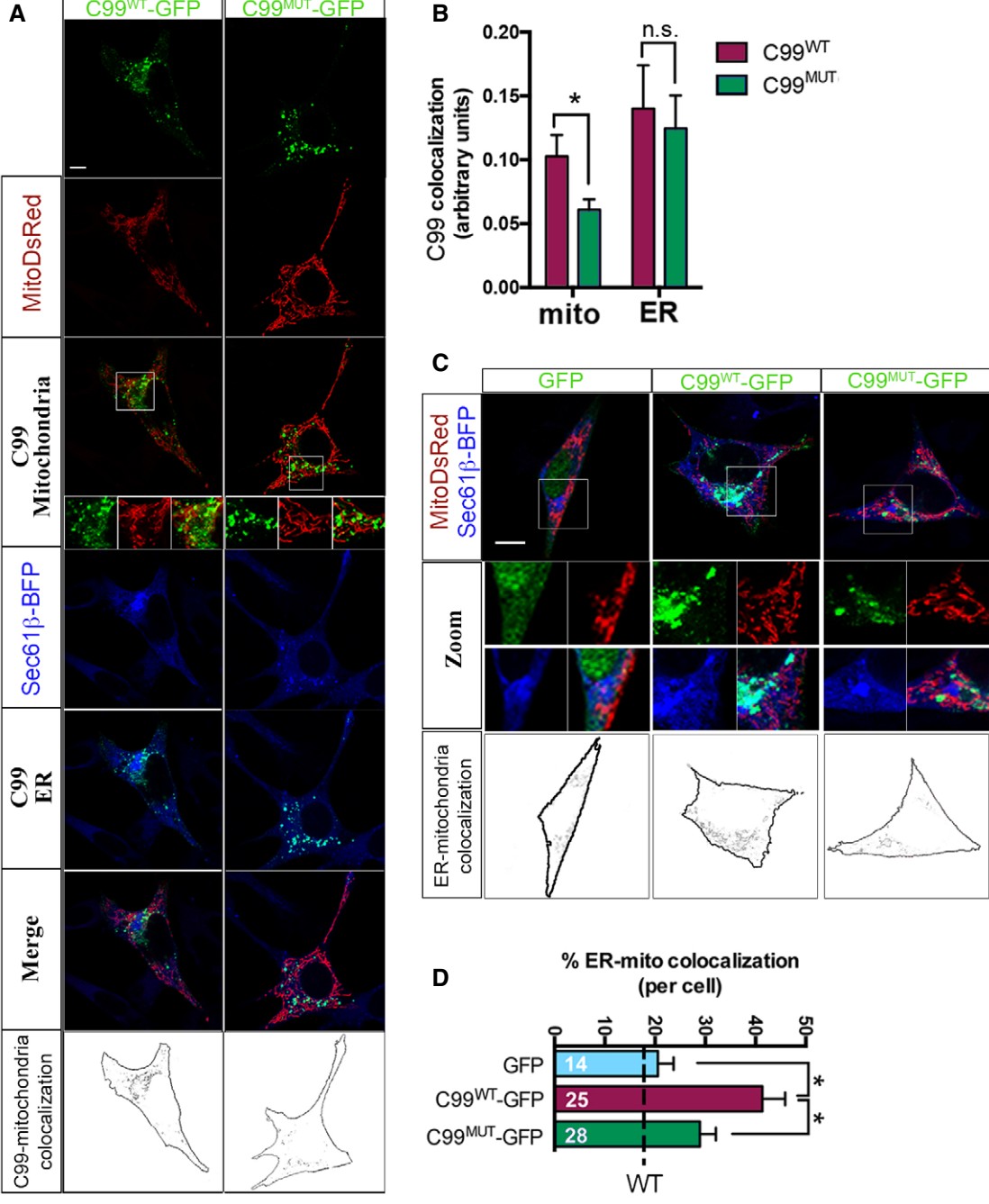

**Figure 4.**

◀

**Figure 4. C99's cholesterol-binding domain facilitates MAM formation and activation.**

A   Representative confocal images of APP-DKO cells expressing GFP-tagged C99$^{WT}$ or C99$^{MUT}$ (in green) and fluorescent markers of mitochondria (MitoDsRed, in red) and ER (Sec61β-BFP, in blue) and treated with DAPT to prevent C99 cleavage. Note the different distribution of C99$^{WT}$ or C99$^{MUT}$ forms. Scale bar = 10 μm. Insets show 5× amplifications of individual (C99 in green, and mitochondria in red) and merged images. Black and white bottom panels represent the areas where C99 and mitochondria signals colocalize. Upon thresholding each channel, a mask for mitochondria was generated and the C99 channel was superimposed on the mitochondria mask, so the positive pixels found in both channels are shown in black.

B   Quantification of the degree of colocalization of C99$^{WT}$ or C99$^{MUT}$ forms with mitochondria or ER measured by ImageJ. Note the significantly decreased colocalization between C99$^{MUT}$ and mitochondria, when compared to that of C99$^{WT}$. Unpaired *t*-test. 20–30 cells/condition from 3 independent experiments and 4–5 images/experiment; *$P < 0.05$; n.s., not significant.

C   Representative confocal images of APP-DKO cells expressing GFP-tagged C99$^{WT}$ or C99$^{MUT}$ and ER and mitochondria markers as in (A) (upper panel). Individual zoomed single channels (middle panel) as well as areas of ER–mitochondria colocalization (bottom panel) are shown for each condition. Scale bar = 10 μm.

D   Quantification of ER–mitochondria colocalization per cell analyzed. Upon thresholding each channel, a mask for ER was generated and the mitochondria channel was superimposed on the ER mask, so the positive pixels found in both channels were quantified and referenced over the total area of the ER mask. The number of cells analyzed is indicated in white text within each column. Levels of ER–mitochondria colocalization in APP-WT cells are shown for reference as a dashed line. EV, empty vector. One-way ANOVA (*n* = 3 independent experiments and 5–8 images/experiment; *$P < 0.05$).

Source data are available online for this figure.

To test this idea, we measured MAM activity and ER–mitochondria crosstalk by quantification of phosphatidylserine (PS) synthesis and its conversion to phosphatidylethanolamine (PE), a known MAM-resident function (Vance, 2014; Montesinos *et al*, 2020). MAM activity was significantly increased in APP-DKO cells expressing C99-$^{WT}$ compared to control conditions (as shown previously, Pera *et al*, 2017); in contrast, APP-DKO cells expressing C99-$^{MUT}$ failed to display significant MAM upregulation. Taken together, these results suggest that C99 binding to cholesterol is important not only for its localization to MAM, but also for the proper formation and activation of MAM itself.

### The cholesterol-binding domain of C99 is required for the formation of MAM domains in the ER

As a lipid raft, MAM (Area-Gomez *et al*, 2012) is a transient functional membrane domain formed by local increases in cholesterol. MAM formation is mediated by peptides with CBDs capable of recruiting cholesterol until it coalesces into a rigid, lipid-ordered domain (Epand *et al*, 2006). Thus, in light of our results, we hypothesized that C99, when delivered to the ER, docks to cholesterol via its CBD, thereby helping form MAM domains. To test this, we decided to analyze C99's affinity for cholesterol using a Click Chemistry approach (Fig 5A). PhotoClick cholesterol is a cholesterol analog conjugated to a photoreactive alkyne that faithfully mimics native cholesterol and can serve as a tool to determine protein affinity for cholesterol (Hulce *et al*, 2013). We incubated APP-DKO cells transiently expressing C99$^{WT}$ or C99$^{MUT}$ with this *trans*-sterol probe. Subcellular fractions from these cell models were then conjugated to an azide-biotin tag by Click Chemistry, followed by a pull-down assay using streptavidin beads (Fig 5A and Appendix Fig S1A–C). As a proof of principle, we were able to detect increased levels of C99$^{WT}$ bound to cholesterol in MAM fractions from PS-DKO cells (Appendix Fig S1D).

The levels of PhotoClick cholesterol in homogenates of APP-DKO cells expressing C99$^{WT}$ or C99$^{MUT}$ were comparable (Fig 5B and C). However, pull-down of the added PhotoClick cholesterol revealed that the amount of C99$^{MUT}$ bound to this lipid was significantly reduced when compared to that of C99$^{WT}$ (Fig 5B and C), confirming that, as reported (Barrett *et al*, 2012), the C99 residues G$_{700}$, I$_{703}$, and G$_{704}$ are crucial for C99 cholesterol binding. Moreover, cells transfected with C99$^{MUT}$ exhibited significant reductions in the

amount of PhotoClick cholesterol that trafficked to MAM (Fig 5B and D). Similarly, pull-down of the PhotoClick cholesterol present in MAM fractions from cells transfected with C99$^{MUT}$ displayed lower levels of bound C99 (Fig 5C and Appendix Fig S1E). In support of this, lipidomics analysis of isolated MAM fractions from these cell models showed reduced levels of cholesterol in cells expressing C99$^{MUT}$ when compared to controls (Fig 5E).

Consistently, transfection of APP-DKO cells with C99$^{WT}$ constructs resulted in the upregulation of cholesterol uptake (Fig 5F), cholesterol esterification by ACAT1 (Fig 5G), increased cholesterol efflux (Fig EV4B), and the concomitant activation of SMase(s) (Fig EV4C), whereas cells transfected with C99$^{MUT}$ showed no differences in these parameters when compared to controls. In agreement with these data, APP-DKO cells expressing C99$^{WT}$ showed a higher ratio of CE:FC (Fig 5H) and a significant increase in lipid droplets (Fig EV4D) when compared to its C99$^{MUT}$ counterpart and to relevant controls.

Altogether, our results suggest that C99's capacity to recruit and cluster cholesterol in the ER triggers the formation and activation of MAM domains. Moreover, accumulation of uncleaved C99 in AD cells induces the continuous turnover of MAM domains by activating the uptake and internalization of cholesterol and its trafficking to the ER (Fig 6).

## Discussion

In previous reports, we and others found that γ-secretase activity is localized in the ER but is not homogeneously distributed throughout this compartment; rather, γ-secretase functionality is enriched in MAM domains, a transient DRM within the ER (Area-Gomez *et al*, 2009; Schreiner *et al*, 2015). Moreover, alterations in γ-secretase activity result in the functional upregulation of this ER subregion and in increased ER–mitochondria apposition (Area-Gomez *et al*, 2012; Hedskog *et al*, 2013).

Recently, we showed that the γ-secretase substrate, C99, is predominantly localized to MAM. Furthermore, we observed that alterations in γ-secretase activity induce an accumulation of this APP fragment in MAM membranes, causing the upregulation of MAM activities—such as cholesterol esterification and sphingolipid turnover—as well as mitochondrial dysfunction (Pera *et al*, 2017). We now show that these phenotypes are consequences of the

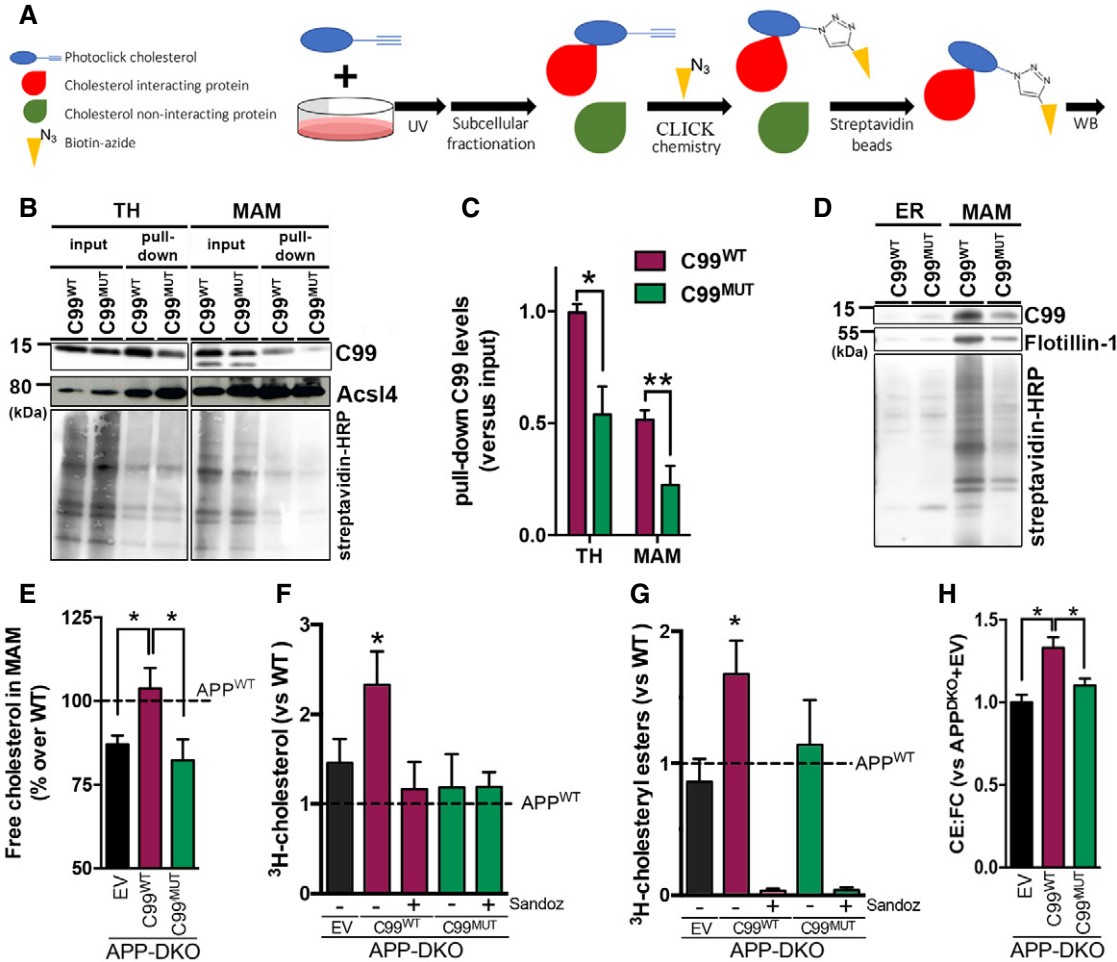

**Figure 5. C99's cholesterol-binding domain is necessary for cholesterol trafficking to MAM.**

A   Scheme of PhotoClick cholesterol methodology to detect C99 interaction with cholesterol.

B   Representative immunoblot to reveal C99 levels in total homogenate (TH) and MAM fractions of APP-DKO cells expressing C99WT or C99MUT before (input) and after cholesterol pull-down. Acsl4 was used as a MAM marker. Streptavidin-HRP was used to detect total biotinylation (biotin conjugated to PhotoClick cholesterol).

C   Quantification of pulled-down C99 levels vs. input from the experiment in (B). Two-way repeated measures ANOVA (Fraction, Mutation) ($n = 5$; *$P < 0.05$, **$P < 0.01$).

D   Immunoblot showing the levels of pulled-down PhotoClick Cholesterol (streptavidin-HRP) in ER and MAM fractions. Note how the levels of pulled-down C99 in the ER are negligible when compared to those from MAM. Isolated MAM and ER fractions were assessed by Western blot (shown in Appendix Fig S1E).

E   Quantification of free cholesterol levels (FC) analyzed by lipidomics after subcellular fractionation to obtain MAM from the indicated cells. Lipid units are represented as molar mass over total moles of lipids analyzed (mol%). Graphs represent percentage over controls. Dashed line represents control levels. One-way ANOVA ($n = 4$; *$P < 0.05$).

F, G   Quantification of cholesterol uptake and esterification in the indicated cells was measured by 4 h incubation with ³H-cholesterol and subsequent analysis of radiolabel incorporation. The dashed line indicates control levels. Graphs represent fold change over controls. Treatment with Sandoz 58-035, a specific ACAT1 inhibitor, caused a ~95% reduction in cholesterol esterification. EV, empty vector. One-sample *t*-test ($n = 3$–4; *$P < 0.05$).

H   Ratio of cholesteryl esters:free cholesterol (CE:FC) in the indicated cells. One-way ANOVA ($n = 4$; *$P < 0.05$).

Source data are available online for this figure.

continuous uptake of extracellular cholesterol and its delivery to MAM, provoked by increased levels of C99 in AD cells. Altogether, our data suggest a pathogenic role for elevations in C99 in AD, via upregulation of cholesterol trafficking and MAM activity, and the subsequent disruption of cellular lipid homeostasis.

Our data support a model (Fig 6) in which C99 at MAM induces the uptake and transport of cholesterol from the PM to the ER (Litvinov *et al*, 2018), via an as-yet-unknown mechanism, which results in the inhibition of the SREBP2-regulated pathway(s) (Brown &

Goldstein, 1999). We propose that, under normal circumstances (Fig 6, upper panel), C99—via its CBD—causes cholesterol to cluster in the ER, resulting in the formation of MAM domains. As this MAM-cholesterol pool expands it activates the hydrolysis of sphingomyelin by SMases, exposing cholesterol to ACAT1 for esterification, decreasing the concentration of membrane-bound cholesterol, and resulting in the dissolution of the lipid raft (Chang *et al*, 2006). In this way, C99 promotes a self-regulating feedback loop to help maintain intracellular cholesterol levels. Under this

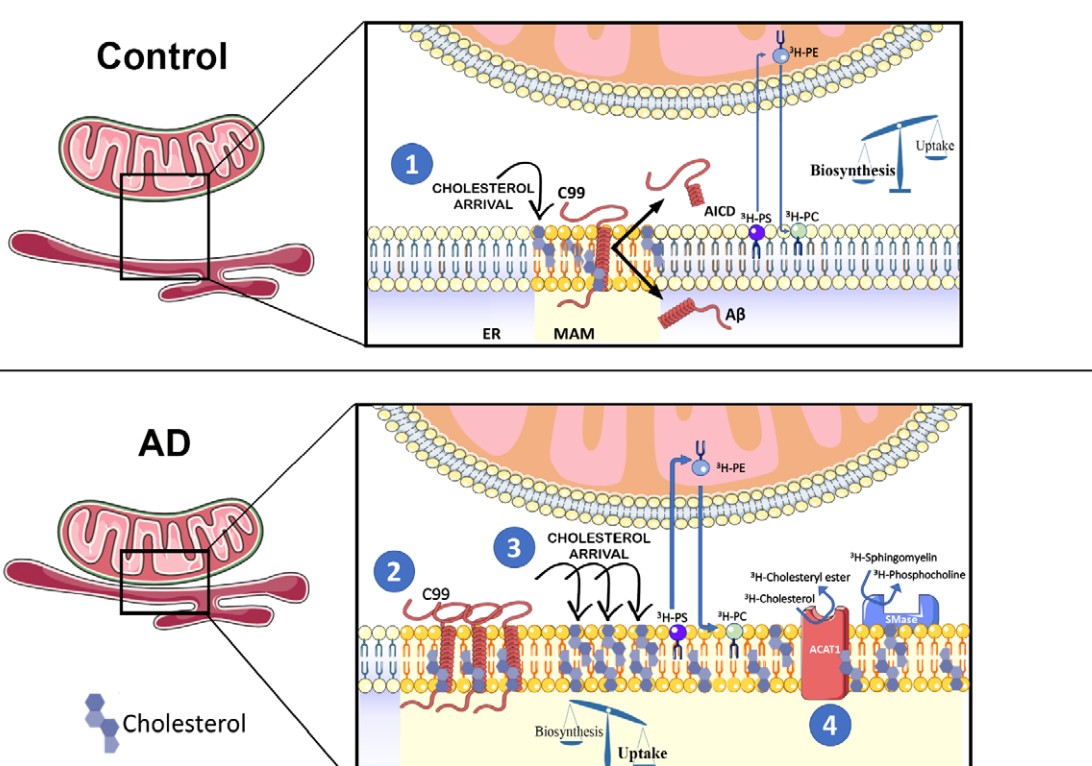

**Figure 6. Schematic representation of the potential role of C99 in the regulation of cholesterol trafficking, and its relevance to AD.**

By means of its affinity for cholesterol, uncleaved C99 at the ER induces the uptake and retrograde transport of cholesterol from the PM to the ER, resulting in the formation of a lipid raft domain, or MAM. These C99-dependent lipid rafts passively segregate and organize lipid-binding proteins, thereby facilitating their interaction and the regulation of specific signaling pathways. Failure to cleave C99 completely would result in a futile cycle of continuous uptake of extracellular cholesterol and its mobilization from the PM to the ER, resulting in the upregulation of MAM formation and activation, which in turn would cause the upregulation of SMases, ACAT activity, and LD deposition. Closing the cycle, this accumulation of cholesterol in membranes also induces APP internalization and its interaction and cleavage by BACE1, and the downregulation of α-secretase activity.

point of view, the failure to cleave C99 (Shen & Kelleher, 2007; Fig 6, lower panel) would result in a futile cycle of continuous uptake of cholesterol and its mobilization from the PM to the ER, resulting in the upregulation of MAM formation and activation, which in turn would cause the upregulation of SMases, ACAT1 activity, and lipid droplet deposition. Closing the cycle, this accumulation of cholesterol in the PM would also induce APP internalization and its interaction with, and cleavage by, BACE1 (Cossec *et al*, 2010; DelBove *et al*, 2019), and the downregulation of α-secretase activity (Wang *et al*, 2014). Interestingly, increases in exogenous cholesterol can mimic this scenario, resulting in APP internalization and elevations in C99 (Marquer *et al*, 2011), increases in the $A\beta_{42:40}$ ratio (Marquer *et al*, 2014), Tau phosphorylation (van der Kant *et al*, 2019), and hippocampal atrophy and cognitive impairment (Djelti *et al*, 2015).

Our results are thus in support of a central role for C99 in the pathogenesis of AD. In the AD brain, there is a substantial increase in the production of amyloid, which originates from elevated levels of C99 fragments (Lauritzen *et al*, 2012). Thus, the buildup of C99 in AD could be considered an early pathological hallmark that may elicit many of the molecular symptoms of the disease, including endosomal dysfunction (Jiang *et al*, 2010), cognitive impairment, and hippocampal degeneration (Lauritzen *et al*, 2012). Interestingly,

others have found that changes in C99, rather than in Aβ or AICD, could be behind some of the symptoms of dementia (Tamayev *et al*, 2012; Pulina *et al*, 2020). In light of these reports and our own data, we believe that C99 toxicity in AD is mediated by its role in cholesterol metabolism.

Consistent with previous data (Pierrot *et al*, 2013), our results show that the buildup of C99 in MAM correlates with significant decreases in HMGCR- and SREBP2-regulated pathways such as the *de novo* synthesis of cholesterol, in an Aβ- and AICD-independent fashion. Thus, while the specific mechanism still needs to be elucidated, these data suggest that the accumulation of C99 prevents SREBP2 activation, impeding its function as a transcription factor (Brown & Goldstein, 1999). We note that one of the SREBP2-regulated genes is the LDL receptor, whose expression would also be reduced by SREBP2 downregulation in a feedback mechanism to control cholesterol levels (Brown & Goldstein, 1999). Paradoxically, our results also reveal that cholesterol uptake is highly induced in γ-secretase-deficient cells in which C99 levels are increased. The intriguing failure of a negative feedback mechanism to downregulate C99-mediated cholesterol import suggests that the continuous uptake of cholesterol might occur through one of the SREBP2-independent cholesterol receptors expressed in the cell (Makar *et al*, 2000; Bindesboll *et al*, 2020).

                                              

As mentioned above, when in excess, "accessible" cholesterol in the PM cholesterol pool will be proportionally transported to the ER cholesterol pool, where it will trigger feedback responses to maintain homeostasis (Das *et al*, 2014; Infante & Radhakrishnan, 2017; Litvinov *et al*, 2018). It has been suggested that this trafficking is regulated by cholesterol-sensing proteins and/or a specific cholesterol-sensing membrane domain in the ER associated with ACAT1 (Lange *et al*, 1999). Based on previous studies (Beel *et al*, 2008) and the results presented here, we propose that C99, via its CBD (Barrett *et al*, 2012), acts as such a cholesterol-sensing protein and that MAM acts as a signaling platform in the regulation of cholesterol homeostasis. Thus, it is possible that, via this affinity domain, the accumulation of C99 generates the cholesterol-rich areas needed for its cleavage by the γ-secretase complex (Wahrle *et al*, 2002). Hence, in the context of deficient γ-secretase activity, uncleaved C99 will continue to recruit cholesterol to MAM, which helps explain the upregulation in MAM activity and ER–mitochondria connectivity found in cells from AD patients (Area-Gomez *et al*, 2012; Hedskog *et al*, 2013). In agreement with this idea, C99^MUT that is defective in cholesterol binding failed to promote the upregulation of MAM functionality.

C99 is not alone in harboring a cholesterol-binding domain with the capacity to modulate the formation and activation of lipid raft domains. This has also been shown for the sigma-1 receptor, a MAM-resident protein (Hayashi & Su, 2007) that, via its capacity to bind cholesterol, triggers the remodeling of lipid rafts and the regulation of the signaling molecules localized therein (Palmer *et al*,

2007). Thus, such cholesterol-sensing proteins might represent a natural mechanism of regulating dynamic lipid rafts like MAM domains.

In summary, we propose a potential mechanism to explain the fundamental role of cholesterol in AD, underscored by the multiple genetic studies that have identified polymorphisms in genes related to cholesterol metabolism and the incidence of AD (Dong *et al*, 2017), including, most prominently, APOE. Moreover, our data help clarify the interdependence between cholesterol and APP metabolism (Marquer *et al*, 2011; DelBove *et al*, 2019) and the controversial association between cholesterol levels and AD (Wood *et al*, 2014), which we believe may be rooted in the fact that defects in neuronal transmission could be caused by alterations in the distribution of subcellular cholesterol (DelBove *et al*, 2019) rather than in overall changes in cellular cholesterol concentration.

Taken together, we propose a model in which C99 accumulation and increased cholesterol uptake occur early in the pathogenesis of AD. Such a model would help create a framework to understand not only the role of cholesterol as both a cause and a consequence in the pathogenesis of AD, but also the participation of many genetic loci associated with lipid metabolism, and specifically cholesterol regulation, in the pathogenesis of AD. In addition, this model supports the idea that the APP C-terminal fragment acts as a cholesterol-sensing protein in the membrane (Beel *et al*, 2008), whose cleavage regulates lipid homeostasis in the cell, coordinating the lipid composition of the PM and the intracellular ER lipid-sensing platform, namely MAM.

# Materials and Methods

### Reagents and Tools table

| Reagents | Supplier | Reference |
|---|---|---|
| Biotin-azide | Click Chemistry Tools | 1167 |
| BSA (fatty acid free) | Sigma-Aldrich | A3803 |
| Cholesterol, [1,2-3H(N)]-, 1 mCi | Perkin Elmer | NET139001MC |
| DAPI | Thermofisher | D1306 |
| DAPT | Sigma-Aldrich | D5942 |
| Desipramine | Sigma-Aldrich | D3900 |
| Filipin | Sigma-Aldrich | F9765 |
| Fluoromount-G™ | Thermofisher | 00-4958-02 |
| GW4896 | Sigma-Aldrich | D1692 |
| L-[3H(G)]-Serine (5 mCi) | Perkin Elmer | NET248005MC |
| LipidTOX™ Green Neutral Lipid Stain | Thermofisher | H34475 |
| LipidTOX™ Red Neutral Lipid Stain | Thermofisher | H34476 |
| Lipofectamine™ 2000 Transfection Reagent | Thermofisher | 11668-027 |
| Methyl-β-cyclodextrin | Sigma-Aldrich | C455 |
| NBD-cholesterol | Thermofisher | N1148 |
| NucRed staining | Thermofisher | R37106 |
| PhotoClick Cholesterol | Avanti Lipids | 700174 |
| Sandoz 58-035 | Sigma-Aldrich | S9318 |
| Sphingomyelin, [choline Methyl-3H]-, 10 μCi | Perkin Elmer | NET1134010UC |

**Reagent and Tools table** (continued)

| Reagents | Supplier | Reference |
|---|---|---|
| Streptavidin (Sepharose® Bead Conjugate) | Cell Signaling | 3419 |
| Streptavidin-HRP | Abcam | ab7403 |
| TLC phospholipid markers | Sigma-Aldrich | P3817 |
| Tris(2-carboxyethyl)phosphine | Sigma-Aldrich | C4706 |
| Tris[(1-benzyl-1H-1,2,3-triazol-4-yl)methyl]amine | Sigma-Aldrich | 678937 |
| XT MES buffer | Bio-Rad | 1610789 |

| Antibodies | Supplier | Reference |
|---|---|---|
| ACSL4 | Sigma-Aldrich | SAB2701949 |
| Actin beta | Sigma-Aldrich | A5441 |
| Alexa Fluor 594 anti-rabbit | Thermofisher | A32740 |
| APP C-terminal | Sigma-Aldrich | A8717 |
| APP C99 (6E10) | Covance | SIG-39320 |
| ATP5A1 | Invitrogen | 459240 |
| Erlin-2 | Cell Signaling | 2959 |
| ERp72 | Cell Signaling | D70D12 |
| Flotilin 1 | Sigma-Aldrich | F1180 |
| GM130 | Thermofisher | PA1-077 |
| IgG mouse-HRP | Sigma-Aldrich | GENA931V |
| IgG rabbit-HRP | Sigma-Aldrich | GENA934V |
| LAMP1 | Thermofisher | 9091S |
| Rab5a | Cell Signaling | 3547 |
| Rab7 | Cell Signaling | 9367 |
| Sec61β | Thermofisher | PA3-015 |
| SREBP2 | Abcam | ab30682 |
| Tubulin | Sigma-Aldrich | T4026 |

## Methods and Protocols

### Cells

WT and PSEN1/2-DKO (called PS-DKO) mouse embryonic fibroblasts (MEFs) were provided by Dr. Bart De Strooper (University of Leuven). APP$^{WT}$ and APP/APLP2-KO (called APP-DKO) mouse embryonic fibroblasts (MEFs) (Herms *et al*, 2004) were a kind gift from Dr. Huaxi Xu (Sanford Burnham Institute). SH-SY5Y and Neuro-2a cell lines were obtained from the American Type Culture Collection. AD and control fibroblasts were obtained from the Coriell Institute for Medical Research (Camden, NJ, USA). Other PS1-mutant FAD cells were kind gifts from Dr. Gary E. Gibson (Cornell University).

Human induced pluripotent stem cells (hiPSCs) in which the APP$^{V717I}$ (London) mutation was knocked into both alleles of the control IMR90 cl.4 iPSC line (WiCell) (Yu *et al*, 2007, Yu *et al*, 2009; Chen *et al*, 2011; Hu *et al*, 2011) were generated by Dr. Andrew Sproul's laboratory, as described previously for the heterozygous knock-in of this mutation (Sun *et al*, 2019). APP$^{V717I}$ knock-ins and the isogenic parent line were maintained feeder-free in StemFlex media (Life) and Cultrex substrate (Biotechne). Cortical neurons from C57BL6 WT or PS1(M146V)-KI mice were cultured from P0 pups as reported (Skaper & Facci, 2018) under the approval of the

Institutional Animal Care and Use Committee of the Columbia University Irving Medical Center.

### Plasmid constructs and transfections

Plasmids were constructed using standard techniques. Construction of C99, AICD, and C99-GFP (a kind gift from Dr. Albert Lleo) plasmids was described in Pera *et al* (2017).

C99$^{MUT}$ was constructed using a megaprimer method. First, a PCR was performed using the pCAX APP-695 plasmid (Addgene #30137) as a template, forward primer: 5′-GTTCAAACAAAG**C**TG-CAATC**GC**TG**C**ACTCATGGTGGG-3′ and reverse primer: 5′-CCC<u>GGA TCC</u>AAGCTTCTAGTTCTGCATCTGCTCAAAGAACTTG-3′, to obtain a megaprimer with the mutations (in bold). The product of this PCR was used as a reverse primer, with forward primer: 5′-ATAC-G<u>AAGCTT</u>GCAGAATTCCGACATGACTCA-3′ and the pCAX APP-695 plasmid as a template. The final PCR product was digested using HindIII/BamHI (restriction sites underlined) and ligated into pGFP-N3. C99$^{MUT}$-GFP was generated by PCR using the C99$^{MUT}$ plasmid as a template, the forward primer: 5′-ATACG<u>AAGCTT</u>GCAGAATTCC-GACATGACTCA-3′ and the reverse primer: 5′-AGGT<u>GGATCC</u>CGTT CTGCATCTGCTCAAAGAACTTG-3′. The PCR product was ligated into the C99-GFP plasmid previously digested with HindIII/BamHI.

C83 was amplified from the pCAX APP-695 plasmid (Addgene #30137) using the forward primer: 5′-CCCGAATTCATGTTGG TGTTCTTTGCAGAAGATGTG-3′ and the reverse primer: 5′-CTAAAGCTTCTAGTTCTGCATCTGCTCAAAGAACTTGTAG-3′ and ligated into pGFP-N3 previously digested with EcoRI/HindIII (restriction sites underlined).

All plasmids were verified by restriction analysis and sequencing. Cells were transfected using Lipofectamine™ 2000 Transfection Reagent in serum-free DMEM for 4–6 h according to manufacturer instructions. The transfection and expression efficiency for C99$^{WT}$ and C99$^{MUT}$ were found to be similar (Appendix Fig S1A).

### Aβ$_{42}$ treatment and Aβ$_{40}$ and Aβ$_{42}$ detection

APP-DKO cells were incubated for 16 h with 5 μM of Aβ$_{42}$ oligomers prepared as in Pera *et al* (2017). To detect Aβ$_{40}$ and Aβ$_{42}$ in cortical neuron media, commercial ELISA kits were used following manufacturer instructions (WAKO chemicals ELISA KIT 290-62601 and 294-62501).

### Analysis of ER–mitochondria apposition

Cells were co-transfected with BFP-Sec61β (Addgene, #49154), MitoDsRed (Clontech, #632421), and C99$^{WT}$-GFP or C99$^{MUT}$-GFP at a 1:1:3 ratio. Twelve hours post-transfection, images of triple-transfected cells were acquired, and ER–mitochondria apposition was analyzed as described (Guardia-Laguarta *et al*, 2014). For C99 colocalization over ER or mitochondria, the same approach was applied using the respective signals.

### Subcellular fractionation and sucrose density gradient ultracentrifugation

Purification of ER, crude membranes (CM), and MAM was performed and analyzed as described (Area-Gomez, 2014). 1 mg of CM was incubated at 4°C for 1 h in 0.2% Triton X-100 and loaded onto a continuous sucrose density gradient (10–40%). Upon 16-h centrifugation at 100,000 *g* at 4°C, 24 fractions of 200 μl each were sequentially collected from the top without disturbing the gradient and equal volumes (40 μl) were used for WB.

### Western blotting

For C99 detection, samples were boiled in NuPAGE™ LDS Sample Buffer with 10% β-mercaptoethanol and run in 4–12% Bis-Tris gels (Criterion XT Precast Midi Gels, Bio-Rad) in XT MES buffer. Other proteins were detected using the antibodies listed in Reagents Table.

### Silencing of PSEN1/2

To knock down mouse Presenilin-1 and Presenilin-2 in Neuro-2a cells, shRNAs against *Psen1* (Sigma SASI_Mm01_00048853) and *Psen2* (Sigma SASI_Mm02_00310708) were transiently transfected together, according to the manufacturer's instructions, as in Area-Gomez *et al* (2012) and Chan *et al* (2012). Briefly, cells plated at low confluence were transfected with each shRNA to a final concentration of 30 nM, using Lipofectamine™ 2000 (Invitrogen, 11668-027), at a 1:1 ratio in serum-free DMEM. After 5 h, the medium was changed to DMEM containing 2% FBS and the cells were incubated for 12 h. Successful silencing of the targeted proteins was checked by Western blot.

### Inhibitors

Cells were treated with either 100 nM β-secretase inhibitor IV (BI) or 10 μM DAPT to inhibit β-secretase or γ-secretase, respectively. Inhibition of acid SMase or neutral SMase activities was performed using 10 μM desipramine or 5 μM GW4869, respectively. To inhibit cholesterol esterification, 20 μM 3-[Decyldimethylsilyl]-N-[2-(4-methylphenyl)-1-phenethyl] propanamide (Sandoz 58-035) was used. All drugs were incubated for 12–16 h and DMSO was used as vehicle.

### Cholesterol trafficking and esterification assays

Cholesterol trafficking and esterification were measured as previously described (Pera *et al*, 2017). Cells were incubated for 2 h in serum-free medium to ensure removal of exogenous lipids. 2.5 μCi/ml of $^3$H-cholesterol was added to serum-free DMEM containing 2% FAF-BSA, allowed to equilibrate for at least 30 min at 37°C, and the radiolabeled medium was added to the cells for the indicated periods of time. Cells were then washed and collected in PBS. For some experiments, cells were subjected to subcellular fractionation, removing a small aliquot for protein quantification. Equal protein amounts were used to extract lipids by using three volumes of chloroform:methanol (2:1 *v/v*). After vortexing and centrifugation at 8,000 *g* for 5 min, the organic phase was blown to dryness under nitrogen. Dried lipids were resuspended in 30 μl of chloroform:methanol (2:1 *v/v*) and applied to a TLC plate along with unlabeled standards. A mixture of hexanes/diethyl ether/acetic acid (80:20:1 *v/v/v*) was used as solvent. Iodine-stained bands corresponding to cholesterol and cholesteryl esters were scraped and counted.

### Cholesterol efflux

Cells were incubated with 2.5 μCi/ml of $^3$H-cholesterol prepared as indicated above. One hour after incubation, cells were washed to remove the excess of exogenous radioactive cholesterol and incubated in unlabeled media. After the indicated post-incubation times, media was recovered, briefly centrifuged at 1,500 *g* 5 min to remove any debris, transferred to scintillation vials with 5 ml of Scintiverse BD (Fisher Scientific) and measured in a Scintillation Counter (Tri-Carb 2819TR; Perkin Elmer).

### Phospholipid transfer

Performed as in Area-Gomez *et al* (2012); Area-Gomez (2014); Montesinos *et al* (2020).

### HMGCR activity assay

HMGCR activity was measured following manufacturer's instructions (HMG-CoA Reductase Assay Kit CS1090; Sigma-Aldrich).

### Cholesterol and lipid droplet staining and immunofluorescence

Fluorescent cholesterol analog, NBD-cholesterol (22-(N-(7-Nitrobenz-2-oxa-1,3-Diazol-4-yl)Amino)-23,24-Bisnor-5-Cholen-3β-OI) was used to determine cholesterol uptake.

Filipin staining was performed by incubation with 50 μg/ml of filipin complex in 10% FBS in DPBS for 2 h at room temperature. For neurons, the concentration of filipin was 0.5 mg/ml. After extensive washes, coverslips were mounted with Fluoromount-G™ (Thermo Fisher Scientific) and visualized by confocal fluorescence microscopy using a UV filter set (Leica SP8 confocal microscope; 340–380 nm excitation, 40 nm dichroic, 430-nm long pass filter).

For immunofluorescence, prior to filipin staining, cells were permeabilized using saponin 0.05%, BSA 5% in PBS for at least 45 min, extensively washed and incubated with primary antibodies, as listed in Reagents Table, for 1 h at room temperature. After extensive washes, 1 h incubation with secondary antibodies conjugated to Alexa fluorophores was performed before filipin staining.

Staining of lipid droplets was performed using HCS LipidTox™ Deep Green or Red neutral lipid stain according to manufacturer instructions. Staining was quantified using ImageJ. Reported fluorescence intensity represents the product of the intensity and the area covered by the fluorescent signal above background in every cell examined. For some experiments, the number of cells containing lipid droplets or filipin-positive punctae was counted and reported. In some experiments, DAPI or NucRed® was used to visualize nuclei.

### Sphingomyelinase activity

Sphingomyelinase activity was assayed as previously described (Pera *et al*, 2017). 100 μg protein was assayed in 100 mM of the appropriate buffer (Tris/glycine for pH 7.0–9.0 or sodium acetate for pH 4.0–5.0), 1.55 mM Triton X-100, 0.025% BSA, 1 mM MgCl$_2$, and 400 μM bovine brain sphingomyelin spiked with 22,000 dpm of [$^3$H]-bovine sphingomyelin (1 nCi/sample). Reactions were carried out in borosilicate glass culture tubes at 37°C overnight, followed by quenching with 1.2 ml ice-cold 10% trichloroacetic acid, incubation at 4°C for 30 min, and centrifugation at 380 *g* at 4°C for 20 min. 1 ml supernatant was transferred to clean tubes, 1 ml ether was added, the mixture vortexed, and centrifuged at 380 *g* for 5 min. 800 μl of the bottom phase was transferred to scintillation vials with 5 ml of Scintiverse BD (Fisher Scientific) and measured in a Scintillation Counter (Tri-Carb 2819TR, Perkin Elmer).

### Lipidomics analysis

Lipids were extracted from equal amounts of material (30 μg protein/sample). Lipid extracts were prepared via chloroform–methanol extraction, spiked with appropriate internal standards, and analyzed using a 6490 Triple Quadrupole LC/MS system (Agilent Technologies, Santa Clara, CA) as described previously (Chan *et al*, 2012). Cholesterol and cholesteryl esters were separated with normal-phase HPLC using an Agilent Zorbax Rx-Sil column (inner diameter 2.1 Å ~100 mm) under the following conditions: mobile phase A (chloroform:methanol:1 M ammonium hydroxide, 89.9:10:0.1, *v/v/v*) and mobile phase B (chloroform:methanol:water:ammonium hydroxide, 55:39.9:5:0.1, *v/v/v/v*); 95% A for 2 min, linear gradient to 30% A over 18 min and held for 3 min, and linear gradient to 95% A over 2 min and held for 6 min. Quantification of lipid species was accomplished using multiple reaction monitoring (MRM) transitions that were developed in earlier studies (Chan *et al*, 2012) in conjunction with referencing of appropriate internal standards. Values are represented as mole fraction with respect to total lipid (% molarity). For this, lipid mass of any specific lipid is normalized by the total mass of all lipids measured (Chan *et al*, 2012).

### PhotoClick cholesterol assay

To study cholesterol interaction with C99, a method developed by Hulce *et al* (2013) was used. Briefly, 24 h after transfection, cells were incubated in serum-free medium for 2 h to remove all exogenous lipids. After that, 5 μM PhotoClick cholesterol (Hex-5′-ynyl 3β-hydroxy-6-diazirinyl-5α-cholan-24-oate), previously complexed with an aqueous saturated solution of MβCD (38 mM), was added to the cells and incubated for 4 h. Upon washes with DPBS, PhotoClick cholesterol was crosslinked under 365 nm-UV (0.75 J/cm$^2$, UVC 500 Ultraviolet Crosslinker; Amersham Biosciences), washed again, collected, and used for subcellular fractionation as previously described. 500 μg (adjusted to a final 150 μl volume with PBS supplemented with protease inhibitors) of TH, ER, and MAM fractions were briefly sonicated and subjected to click chemistry by addition of 500 μM biotin-azide, 100 μM Tris[(1-benzyl-1H-1,2,3-triazol-4-yl)methyl]amine (TBTA), 1 mM CuSO$_4$, and 1 mM Tris(2-carboxyethyl)phosphine (TCEP) and incubation for 15 min at room temperature in the dark. Then, samples were diluted in 50 mM Tris pH 7.4 with protease inhibitors (removing an aliquot as input) and incubated overnight under rotation at 4°C with streptavidin beads. Upon several washes with Tris 50 mM pH 7.4, beads were collected by centrifugation at 2,000 rpm for 1 min, boiled with NuPAGE™ LDS Sample Buffer (1×) 95°C for 5 min and used for immunodetection. A diagram is presented in Fig 5A.

### Statistical analysis

Data represent mean ± SEM. All averages are the result of three or more independent experiments, carried out at different times with different sets of samples. Historical and preliminary data were used to ensure adequate power and to limit the background associated with each assay (SigmaPlot 12.0). Data distribution was assumed to be normal. The statistical analysis was performed using GraphPad Prism v5.01 (GraphPad Software Inc., CA, USA). Brown–Forsythe test was used to compare variance between groups. Statistical significance was determined by either two-tailed *t*-test, one-way ANOVA, or (repeated measures) two-way ANOVA, followed by the Newman–Keuls' post hoc test as reported in each figure legend. Greenhouse–Geisser correction was used for repeated measures one-way ANOVA comparison when variance between groups was statistically different, as stated. Values of $P < 0.05$ were considered statistically significant. The investigators were not blinded when quantifying imaging experiments.

# Data availability

This study includes no data deposited in external repositories.

**Expanded View** for this article is available online.

## Acknowledgements

We thank Dr. Huaxi Xu (Sanford Burnham Institute) for the APP-DKO cells, and Drs. Ismael Santa-Maria and Francesca Bartolini for their valuable comments and assistance with AD animal models. We also thank Maria Kaufman for technical assistance with the APP$^{V717I}$ iPSCs. This work was supported by the U.S. National Institutes of Health (K01-AG045335 and R01-AG056387-01 to E.A.-G.), Alzheimer's Association (AARF-19-614721 to J.M.), Alzheimer's Disease Research Center (ADRC) pilot grant (to J.M.), the Michael J. Fox and the Leir Foundations (to C.G.-L.), the Henry and Marilyn Taub Foundation (A.A.S.), and the National Defense Science and Engineering Graduate Fellowship (FA9550-11-C-0028 to R.R.A.).

## Author contributions

EA-G conceived the project and designed and interpreted most of the experiments. JM and EA-G wrote the paper. MP, DL, JM, CG-L, RRA, IS, and KRV performed most of the experiments and edited the manuscript. YX and TDY performed the lipidomics analysis. S-YK and AMS generated APP$^{V717I}$ homozygous knock-in iPSC lines under the guidance of AAS, who also edited the manuscript.

## Conflict of interest

The authors declare that they have no conflict of interest.

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
