## [Review Process File · The EMBO Journal]

THE C99 FRAGMENT OF APP REGULATES CHOLESTEROL TRAFFICKING

Jorge Montesinos, Marta Pera, Delfina Larrea, Cristina Guardia-Laguarta, Rishi Agrawal, Kevin Velasco, Taekyung Yun, Irina Stravrovskaya, Yimeng Xu, So Yeon Koo, Amanda Snead, Andrew Sproul and Estela Area-Gomez

DOI: [10.15252/embj.2019103791](https://doi.org/10.15252/embj.2019103791)

Corresponding author(s): Estela Area-Gomez (eag2118@cumc.columbia.edu)

Review Timeline:

Submission Date:	20th Oct 19
Editorial Decision:	6th Dec 19
Revision Received:	12th May 20
Editorial Decision:	3rd Jul 20
Revision Received:	14th Jul 20
Accepted:	22nd Jul 20

Editor: Karin Dumstreit

Transaction Report:

Dear Estela,

Thank you for submitting your manuscript to The EMBO Journal. Your study has now been seen by three referees and their comments are provided below.

While the referees appreciate the importance of the topic and see that the findings add new insight, they also find that the analysis is not taken far enough in order to consider publication here. As you can see from the comments below, the referees find that further data is needed in order to support that key conclusions reported and that the analysis needs to be extended. Further work in neuronal cells and a more relevant AD model system is also needed. Given the raised concerns and as it is unclear if they can be resolved, I am afraid that I see no other choice but to reject the manuscript at this stage.

Should further work allow you to extend the findings along the lines as indicated by the referees then I am open to consider a new submission on this topic. I should point out that for re-submissions that we consider novelty at time of submission and might involve new referee(s) if needed. If you are interested in a resubmission please contact me beforehand to discuss the extent of the revisions.

For the present submission, I am afraid that I can't be more positive on this occasion.

Best Karin

Karin Dumstrei, PhD
Senior Editor
The EMBO Journal

Referee #1:

In this manuscript, authors report that C99, a fragment of APP, plays an important role in cholesterol uptake, and that cholesterol binding domain (CBD) of C99 is crucial for cholesterol delivery to the MAM (mitochondria-associated membranes). These findings might indicate that C99 at MAM regulates lipid homeostasis in Alzheimer's disease. One crucial experiment is the mutation of the C99 CBD that substantiates a role for this portion of the protein in perimitochondrial localization. Authors are advised to name this construct differently: by calling it CBD, authors confuse the reader unnecessarily.

Overall, the study is very interesting and offers further proof for a role of MAMs and lipid trafficking in AD. Conclusions must be strengthened as suggested below.

Major points

1. Authors didn't number figures in the order of appearance in the texts: figure 5 comes prior to figure 4 in the text. Furthermore, the authors performed the same experiment and showed the results in Figure 3A and 5A separately. Authors should follow the guidelines of the journal.

2. On page 6, Authors state "As before (20), incubation with the SMase(s) inhibitors reduced cholesterol esterification by ACAT1 and the accumulation of LDs in PS-DKO cells (Fig. 2A-B), in DAPT-treated SH-SY5Y cells and AD patient fibroblasts (Fig. S2)." Indeed, Fig. 2A & B show that SMase inhibitor treatment leads to increase of neutral lipids in PS-DKO cells; however, it is not clear if the reduction of cholesterol esterification by ACAT1 is involved in the neutral lipids accumulation. This must be addressed
3. Same page "Our data also indicate that this mobilization of cholesterol to ER-MAM has the further effect of reducing de novo cholesterol biosynthesis via inactivation of SREBP2". However, authors did not examine whether the mobilization of cholesterol has an effect on cholesterol biosynthesis, nor whether SREBP2 inactivation is really involved in reduction of cholesterol biosynthesis caused by C99.
4. Page 7, authors describe that C99 binding to cholesterol affects "the proper formation of MAM itself", based on the result shown in Figure 3F. However, the authors show only the change of localization of flotillin. This doesn't always mean MAM deformation. Other direct assays (e.g. proximity between ER and mitochondria, MAM composition etc) must be shown
5. Same page: "Taken together, our results support the idea that the cholesterol binding domain of C99 is necessary to induce cholesterol internalization for the formation of MAM in the ER (34)." Reference 34 reported that depletion of cholesterol in MAM induces reduction in the association between MAM and mitochondria. This seems opposite to what the authors conclude. The authors should cite the reference appropriately and discuss the discrepancy.
6. In Figure 1G, authors show that PS-DKO cells or DAPT-treated cells have more filipin puncta. Localization of these puncta is not thoroughly examined; authors should elucidate whether these puncta are at endolysosome, MAM, ER or another compartment.
7. Supplemental Figure 3A: in C99CBD-GFP-expressing cells, Sec61 β puncta that colocalize with the C99CBD-GFP signal are retrieved. This might be the result of protein aggregation because of high expression levels. If this is not the case, authors should investigate whether the morphology of ER changes overall, or if the Sec61 β localization is affected, by checking localization of other ER markers.
8. in Figure 1I and Figure 4 Authors performed MAM/ER fractionation before cholesterol uptake and pull-down assays. Blots for markers of each fraction should be shown.
9. Supplemental figure 2A-B: Authors show images and quantifications of LipidTox staining in SH-SY5Y cells. From the quantification, LipidTox staining in DAPT treated condition shall be 10X than in other conditions, but the images don't show such a difference. This is a troublesome experiment that must be repeated and corrected
10. In supplemental figure 5A, GFP localization is shown as control, but it seems to localize to some specific organelle like the ER rather than being diffuse in cytosol. The authors should confirm that the "GFP" is not fused with other protein.

Minor points

1. Authors placed a part of the discussion in the Legend of Figure 2E. This should be moved to discussion.
2. The Materials and Methods section contains mistakes that probably derive from some hasty assembly of the manuscript. Authors are advised to carefully re-read the manuscript before submitting it. For example, in the section "Cells", authors cited the reference as PubMed ID, and did not insert it in the reference list. "Reference 65" is not in the list, there are only 48 references. The authors cite reference 23 for detailed protocols, but this is a review article.
3. Please move the detailed statistic information from Figure Legends to supplemental information.

Referee #2:

In this manuscript, Pera and colleagues present a series of evidences that suggest the role of the APP-C99 fragment in the dynamics of cellular cholesterol in fibroblasts, and that conditions accompanied by an increase in this peptide would lead to defects in this pathway and as a consequence to defects in the cell membrane. While the experiments are sound, the conclusions that something similar may occur in the context of Alzheimer's disease, in which associations with genes related to the metabolism / transport of this lipid have been seen, are not supported by the current data.

Critique:

1) The lack of data in neurons is worrying (authors should not argue that Sy5Y cells are neurons), which leads me to question the biological relevance of the results in the context of a CNS disease as it is AD. As the authors certainly know, the plasma membrane of neurons is very different from non-neuronal cells, not only in terms of axonal and dendritic domains but also in the subdomains in each of these domains: i.e. synaptic and non-synaptic, all of which have different lipidic composition and, quite relevant for the argument of this work, present different association with underlying membrane organelles, especially the ER. In addition to the differences in membrane lipid/protein composition and organisation, neurons and fibroblasts differ in their cholesterol synthesis and catabolism activities, especially relevant in the context of age (another relevant variable for AD). Therefore, authors must perform, if not all some of the most critical experiments on AD suitable experimental models (i.e. analysis of intracellular cholesterol content in neurons from PS mutant mice, or directly mutagenised neurons; uptake (and efflux, see below) of labelled cholesterol in neurons from PS mutant mice). The field of AD has progressed so much in recent years that it is necessary to present experiments where biochemical/cell biological mechanistic data in fibroblasts are validated in cells and models closer to the disease.

2) In the introduction, it is stated that: "These higher levels of A β in AD are the consequence of corresponding increases in the cleavage of endocytosed full-length APP by β -secretase to produce the immediate precursor of A β , the 99-aa C-terminal domain of APP (C99), and its subsequent processing by γ -secretase"

Authors need to revise this sentence, as the cause of brain A β accumulation in AD patients is still not clear. It could be a combination of lower clearance or higher production. In familial cases, it seems to be due to the alteration of the carboxypeptidase activity of the gamma-secretase complex due to mutations in presenilin, that triggers the increased production of longer and more toxic Abeta species (therefore, not affecting APP-C99 levels) (Chavez-Gutierrez et al., EMBO J, 2012).

3) In the introduction, it is stated that: "Further linking AD and cholesterol, APP-CTFs processing occurs in lipid rafts (14)"

Authors need to correct this sentence, as processing can occur in other detergent-resistant microdomains (DRM), such as tetraspanin-enriched microdomains (Wakabayashi et al, Nat. Cell. Biol., 2009). A simple solution would be to replace the term "lipid raft" by DRMs.

4) In Figure 1E-F, it is shown that PS-DKO cells contain higher levels of fluorescently-labeled cholesterol, which is attributed to increased uptake.

An alternative to increased uptake is reduced removal from endosomes/MVBs. In light of the argument that C99 increases uptake, authors need to demonstrate that cholesterol efflux from endosomes/MVBs remains unaffected.

5) In this sense, in Figures 2C-F it is shown that cholesterol uptake is not reversed by SMase inhibition in either DAPT-treated WT cells (Fig. 2C-D), or in PS-DKO cells (Fig. 2E-F).

Ceramide triggers budding of exosome vesicles into multivesicular endosomes, which are enriched in cholesterol. Thus, inhibition of SMase could trigger the accumulation of cholesterol.

6) The order of appearance of some figures in the text is altered. The references to the figures in the text jump from Figure 2 to Figure 5: example: "While C99WT showed a perinuclear pattern of colocalization with ER and mitochondria, C99CBD presented a less marked perinuclear localization and a decreased association with mitochondria (Figs. 5A, 5B, and S5A)."

7) The co-localization between C99 wt and mitochondrial and ER markers is not obvious in Figures 3A and Figure 5A. Maybe it would help to show a zoom in Figure 5A.

Referee #3:

The manuscript by Pera et al. investigates how a fragment of APP control cholesterol homeostasis between the ER and the plasma membrane. The manuscript is situated in the control of lipid raft formation, a mechanism that depends on the enrichment of cholesterol under the control of sphingomyelin. The activities and amounts of cholesterol are tightly regulated at the level of the ER via interactions with the plasma membrane and mitochondria at the MAMs.

In the manuscript, Pera et al. provide new data on the processing of APP, whose C99 fragment is generated on intercellular lipid rafts. The authors had previously shown that this processing localizes to MAMs, ER-associated lipid rafts in the proximity of mitochondria.

In this new manuscript, they now provide evidence that C99 is required for the lipid/cholesterol-detoxifying activities of MAMs as one of the functions of this ER domain.

The manuscript currently falls short of looking beyond these functions. However, given the known, well-characterized function of MAMs in multiple signaling events, the investigation of MAM functions should not be limited to lipid homeostasis. Instead, some aspect of other MAM functions should be added to gain a better understanding of what is going on.

Another area of deficiencies is a current lack of link with tethering mechanisms.

At the moment, there are also some inconsistencies in data presentation, i.e., some assays are done for some conditions, but not others.

Nevertheless, this is an important topic that warrants high profile exposure.

Major Points

1. Some MAM signaling functions beyond lipid homeostasis should be investigated to some extent. For instance, calcium transfer from the ER to mitochondria should be shown. This is critical, since we do not know very well how these diverse functions interact with each other.
2. Are tethers affected by C99 and its production? One way to address this could be via analysis of tether complex co-immunoprecipitation or via targeting of tethers to MAMs.

Specific Points:

1. In figure 1, there is inconsistency of data presentation between 1A and 1B. The % molarity is very different between the two. It is also unclear that 1B is total homogenate. Why was this not analyzed for the fractions like 1A?
2. A loading control is missing in Figure S1C.
3. What is the cholesterol uptake in APP DKO and transfectants of C99? Figure 1 just shows PS DKO. We need to understand the relative defects of the cellular models used. This is inconsistent presentation.
4. A description of the filipin staining pattern via co-localization with cellular markers in the PS-DKO is necessary.

5. Supplemental figure 3 should contain zoomed-in areas, similar to Figure 3.
6. The gradient fractionation should contain pan-ER and mitochondrial markers, to allow for judging where the analyzed proteins have fractionated to.
7. It is currently impossible to understand where the APP DKO MAM formation stands relative to wild type MEFs. This control should be added to Figure 5B. Also, would the chemical removal of cholesterol show a similar effect in wild type cells (that would not be observed in DKO)?

Minor concerns:

1. The first paragraph of the Results section is difficult to read.
2. What does the abbreviation CTFs stand for?

** As a service to authors, EMBO Press provides authors with the possibility to transfer a manuscript that one journal cannot offer to publish to another EMBO publication or the open access journal Life Science Alliance launched in partnership between EMBO Press, Rockefeller University Press and Cold Spring Harbor Laboratory Press. The full manuscript and if applicable, reviewers' reports, are automatically sent to the receiving journal to allow for fast handling and a prompt decision on your manuscript. For more details of this service, and to transfer your manuscript please click on Link Not Available. **

Manuscript EMBOJ-2019-103791**Response to reviewers' comments**

We would like to thank the reviewers for their comments and suggestions, most of which have helped make the manuscript stronger. In addition, we apologize for the mistakes in the order of the figures in the previous version. All substantive changes have been highlighted in blue.

Referee#1:

In this manuscript, authors report that C99, a fragment of APP, plays an important role in cholesterol uptake, and that cholesterol binding domain (CBD) of C99 is crucial for cholesterol delivery to the MAM (mitochondria-associated membranes). These findings might indicate that C99 at MAM regulates lipid homeostasis in Alzheimer's disease. One crucial experiment is the mutation of the C99 CBD that substantiates a role for this portion of the protein in perimitochondrial localization. Authors are advised to name this construct differently: by calling it CBD, authors confuse the reader unnecessarily. Overall, the study is very interesting and offers further proof for a role of MAMs and lipid trafficking in AD. Conclusions must be strengthened as suggested below.

Following the referee's recommendation, the mutant version of C99 (formerly termed C99^{CBD}) has been changed to C99^{MUT} in the text and figures.

Major points

1. Authors didn't number figures in the order of appearance in the texts: figure 5 comes prior to figure 4 in the text. Furthermore, the authors performed the same experiment and showed the results in Figure 3A and 5A separately. Authors should follow the guidelines of the journal.

We apologize for this mistake. We have renumbered the figures according to the order of appearance.

2. On page 6, Authors state "As before (20), incubation with the SMase(s) inhibitors reduced cholesterol esterification by ACAT1 and the accumulation of LDs in PS-DKO cells (Fig. 2A-B), in DAPT-treated SH-SY5Y cells and AD patient fibroblasts (Fig. S2)." Indeed, Fig. 2A & B show that SMase inhibitor treatment leads to increase of neutral lipids in PS-DKO cells; however, it is not clear if the reduction of cholesterol esterification by ACAT1 is involved in the neural lipids accumulation. This must be addressed

Fig. 2A shows that treatment with SMase inhibitor(s) Desipramine or GW4869 in PS-DKO cells leads to a reduction in cholesterol esterification and lipid droplet formation. We replicated these results in DAPT-treated SH-SY5Y cells (Fig. S2A) and in cells from FAD patients (Fig. S2B). As a control for ACAT1 contribution to this phenotype, we incubated these same DAPT-treated SH-SY5Y cells with 20 μ M of Sandoz, a specific inhibitor of ACAT1, for 12h. (Fig. S2A)

Our results are also confirmed by the significant reductions in the incorporation of ^3H -cholesterol into ^3H -cholesteryl esters upon treatment with SMase inhibitors (Desipramine and GW4869) in PS-DKO cells (shown below and in Fig. 2, Supp. Fig. 2).

These data together strongly support that, as published before (Das *et al*, 2014; Endapally *et al*, 2019; Slotte & Bierman, 1987, 1988), SMase activation and cholesterol esterification by ACAT1 are well-known, concomitantly activated pathways to detoxify membranes from an excess of cholesterol, leading to neutral lipid and lipid droplet increases in cell models of AD.

Moreover, as shown in figures 2B, 2C and 2D (middle panel), inhibition of SMases does not reduce cholesterol uptake and internalization in these cell models, as shown by staining of endogenous cholesterol by filipin (Figs. 2B and 2C, and middle panel in 2D), and the uptake of fluorescent cholesterol analogs (NBD-cholesterol, Fig 2D-upper panel).

Altogether, these data indicate that cholesterol esterification and SMase activation are consequences, rather than the cause, of the upregulation in cholesterol uptake. As an additional control, we tested whether SMase could affect ACAT1 regulation. As shown below, ACAT1 inhibition was not able to rescue the upregulation in SMase activity in PS-DKO cells.

3. Same page "Our data also indicate that this mobilization of cholesterol to ER-MAM has the further effect of reducing de novo cholesterol biosynthesis via inactivation of SREBP2". However, authors did not examine whether the mobilization of cholesterol has an effect on cholesterol biosynthesis, nor whether SREBP2 inactivation is really involved in reduction of cholesterol biosynthesis caused by C99.

We have changed the text (page 5) to: "HMGCR activity was reduced significantly in PS-DKO MEFs, in WT cells treated with γ -secretase inhibitors (DAPT) (Fig. 1C), and in AD fibroblasts (Fig. S1B), in agreement with previous observations (Pierrot *et al*, 2013). Consistent with this, the levels of both uncleaved and mature SREBP2 protein were significantly reduced in the PS-DKO cells and in WT cells treated with DAPT, compared to WT (Fig. S1C)."

4. Page 7, authors describe that C99 binding to cholesterol affects "the proper formation of MAM itself", based on the result shown in Figure 3F. However, the authors show only the change of localization of flotillin. This doesn't always mean MAM deformation. Other direct assays (e.g. proximity between ER and mitochondria, MAM composition etc) must be shown.

Our data shows that expression of C99^{MUT}, in comparison to C99^{WT}, results in:

- Reduced localization of C99 to MAM (Figs. 3, 4A, Figs, 5B, 5C and 5D).
- Reduced cholesterol levels at MAM (Figs. 5B, 5D and 5E).
- Reduced uptake of cholesterol (Fig. 5F) and cholesterol efflux (Fig. S3E).
- Reduced ER-mitochondria apposition (Figs 4C and 4D).
- Reduced MAM activity as measured by phospholipid transfer (Fig. 4E), ACAT activity (Fig. 5G,5H and S3G), and SMase activity (Supp. Fig. 3F).

All these results strongly confirm that cholesterol binding of C99 is crucial for promoting the physical association of MAM with mitochondria and functional MAM activation, while mutant C99 failed to induce this upregulation. Altogether, our data suggest that, as a lipid raft, decreases in cholesterol delivery and/or defects in cholesterol "clustering" in the ER result in reduced formation of MAM domains.

5. Same page: "Taken together, our results support the idea that the cholesterol binding domain of C99 is necessary to induce cholesterol internalization for the formation of MAM in the ER (34)." Reference 34 reported that depletion of cholesterol in MAM induces reduction in the association between MAM and mitochondria. This seems opposite to what the authors conclude. The authors should cite the reference appropriately and discuss the discrepancy.

We thank the reviewer for this point. We reference Fujimoto et al to clarify that we are not the first group showing that ER-mitochondria connections are modulated by cholesterol. In their manuscript, Fujimoto and colleagues concluded that isolated MAM/mitochondria fractions and cells incubated with methyl- β -cyclodextrin show increases in ER-mitochondria connections. As stated by the reviewer, our conclusions are contradictory. We believe that this discrepancy is due to the different experimental conditions and aims, as well as the distinct effect of methyl- β -cyclodextrin on the plasma membrane versus intracellular membranes.

First, depletion of cholesterol from isolated cellular fractions as shown in Fujimoto et al, while informative, does not account for the activation of cellular counterbalancing mechanisms to replenish the depleted cholesterol. On the other hand, the authors show a similar effect, namely increases in ER-mitochondria contacts, in cultured cells incubated with methyl- β -cyclodextrin. This is an expected effect because methyl- β -cyclodextrin first depletes cholesterol from the plasma membrane (PM), which in turn increases the expression of cholesterol-related genes (SREBP2, HMG-CoA reductase, etc.) to buffer the effects of methyl- β -cyclodextrin. Therefore, while the data is certainly intriguing, we believe that the use of this cholesterol-depleting compound can confound some of the conclusions drawn by Fujimoto et al.

In summary, both manuscripts agree on the relevance of cholesterol in the modulation of ER-mitochondria contacts and the formation of MAM domains. However, given that MAM is a lipid-raft-like domain (defined as detergent-resistant domains, enriched in cholesterol and sphingomyelin), we are in disagreement that decreases in cholesterol concentration would enhance the formation of these functional ER regions. To avoid confusion, we have eliminated this reference from the text.

6. In Figure 1G, authors show that PS-DKO cells or DAPT-treated cells have more filipin puncta. Localization of these puncta is not thoroughly examined; authors should elucidate whether these puncta are at endolysosome, MAM, ER or another compartment.

Following the referee's suggestion, we have analyzed whether the filipin puncta in PS-DKO cells or DAPT-treated WT cells colocalize with markers of different organelles in Supp. Fig. S1K. Our data shows that filipin puncta colocalize mainly with endosomal (Rab7) and lysosomal markers (LAMP1). As expected, some diffuse filipin staining can also be found in the ER (Sec61 β) or Golgi (GM130).

7. Supplemental Figure 3A: in C99CBD-GFP-expressing cells, Sec61 β puncta that colocalize with the C99^{MUT}-GFP signal are retrieved. This might be the result of protein aggregation because of high expression levels. If this is not the case, authors should investigate whether the morphology of ER changes overall, or if the Sec61 β localization is affected, by checking localization of other ER markers.

We thank the reviewer for this suggestion. We have performed new experiments to confirm that C99^{MUT}-GFP does not cause an aberrant morphology of the ER by confocal analysis. We labeled both C99^{WT}-GFP and C99^{MUT}-GFP expressing cells (green) with ER markers (CALR, ERp72 and Sec61 β). As shown below, we have not detected any significant change in ER morphology after expression of WT or mutant forms of C99.

APP-DKO cells expressing either C99^{WT}-GFP or C99^{MUT}-GFP (shown in green), incubated with DAPT to prevent C99 cleavage, were immunostained for the ER markers CALR or ERP72 (in red, left and middle columns), or co-transfected with the ER marker Sec61 β -BFP (shown in red, right columns)

8. in Figure 1I and Figure 4 Authors performed MAM/ER fractionation before cholesterol uptake and pull-down assays. Blots for markers of each fraction should be shown.

We have now included blots probing for specific markers of each subcellular fraction under study in Supp. Fig. S1F (Fractions assayed in Fig. 1D-F) and Supp. Fig S3D (Fractions assays in Figs. 5B and 5D). Erlin-2 was used as MAM marker and ERp72 as ER marker.

9. Supplemental figure 2A-B: Authors show images and quantifications of LipidTox staining in SH-SY5Y cells. From the quantification, LipidTox staining in DAPT treated condition shall be 10X than

in other conditions, but the images don't show such a difference. This is a troublesome experiment that must be repeated and corrected.

We thank the reviewer for this suggestion. We have obtained new confocal images that more accurately reflect the results of the lipid droplet quantification.

10. In supplemental figure 5A, GFP localization is shown as control, but it seems to localize to some specific organelle like the ER rather than being diffuse in cytosol. The authors should confirm that the "GFP" is not fused with other protein.

We have re-confirmed that sequence of our GFP-expressing plasmids and empty vectors is the correct one. We have taken additional confocal images showing the expected diffuse GFP distribution (below) in contrast to Sec61b-BFP signal shown (in red).

APP-DKO cells were transfected with a vector expressing GFP, used as a control in our experiments to analyze C99 cellular distribution by confocal imaging, under the same conditions as explained for that figure. Co-transfection with Sec61b-BFP (shown in red in the right panel) was also assayed.

Minor points

1. Authors placed a part of the discussion in the Legend of Figure 2E. This should be moved to discussion.

The legend of Figure 2E has been edited and moved to the main text.

2. The Materials and Methods section contains mistakes that probably derive from some hasty assembly of the manuscript. Authors are advised to carefully re-read the manuscript before submitting it. For example, in the section "Cells", authors cited the reference as PubMed ID, and did not insert it in the reference list. "Reference 65" is not in the list, there are only 48 references. The authors cite reference 23 for detailed protocols, but this is a review article.

We apologize for these oversights. We have revised our manuscript and references accordingly.

3. Please move the detailed statistic information from Figure Legends to supplemental information.

Statistical information has been moved to a separate section.

Referee #2:

In this manuscript, Pera and colleagues present a series of evidences that suggest the role of the APP-C99 fragment in the dynamics of cellular cholesterol in fibroblasts, and that conditions accompanied by an increase in this peptide would lead to defects in this pathway and as a consequence to defects in the cell membrane. While the experiments are sound, the conclusions that something similar may occur in the context of Alzheimer's disease, in which associations with genes related to the metabolism / transport of this lipid have been seen, are not supported by the current data.

Critique:

1) The lack of data in neurons is worrying (authors should not argue that Sy5Y cells are neurons), which leads me to question the biological relevance of the results in the context of a CNS disease as it is AD. As the authors certainly know, the plasma membrane of neurons is very different from non-neuronal cells, not only in terms of axonal and dendritic domains but also in the subdomains in each of these domains: i.e. synaptic and non-synaptic, all of which have different lipidic composition and, quite relevant for the argument of this work, present different association with underlying membrane organelles, especially the ER. In addition to the differences in membrane lipid/protein composition and organisation, neurons and fibroblasts differ in their cholesterol synthesis and catabolism activities, especially relevant in the context of age (another relevant variable for AD). Therefore, authors must perform, if not all some of the most critical experiments on AD suitable experimental models (i.e. analysis of intracellular cholesterol content in neurons from PS mutant mice, or directly mutagenised neurons; uptake (and efflux, see below) of labelled cholesterol in neurons from PS mutant mice). The field of AD has progressed so much in recent years that it is necessary to present experiments where biochemical/cell biological mechanistic data in fibroblasts are validated in cells and models closer to the disease.

We have performed new assays on cultured cortical neurons explanted from WT mice or PS1^{M146V}-KI, a recognized mouse model of AD (Guo *et al*, 1999). Our data shows that, as in cell models of AD, DAPT-treated WT neurons replicate the upregulation in cholesterol uptake, internalization and subsequent esterification, as measured by incubation with radiolabeled (Fig. 1I) or fluorescent cholesterol (NBD-cholesterol uptake; Fig 1H), filipin staining of endogenous levels of free cholesterol, and visualization of lipid droplets by Lipidtox staining (Fig. 1H). Cortical neurons from PS1^{M146V}-KI mice showed increases in C99 levels (Supp. Fig. S1L and S1M) as shown before (Pera *et al*, 2017), a higher ratio of A β ₄₂/A β ₄₀ (Supp. Fig. S1N), and increased MAM activity [Fig. 1I shows ACAT1 activity and Supp. Fig. 1O shows phospholipid synthesis and transfer between ER and mitochondria]. In support of a role for C99 in the induction of this phenotypes, the above-mentioned alterations in cholesterol metabolism were abrogated by incubation with a BACE inhibitor. Moreover, our data show that these phenotypes are also present in induced pluripotent stem cells (iPSCs) in which a pathogenic mutation in APP (London mutation, APP^{V717I}) was knocked into both alleles using CRISPR/Cas9 (Supp. Figs. S1Q-S).

2) In the introduction, it is stated that: "These higher levels of A β in AD are the consequence of corresponding increases in the cleavage of endocytosed full-length APP by β -secretase to produce the immediate precursor of A β , the 99-aa C-terminal domain of APP (C99), and its subsequent processing by γ -secretase". Authors need to revise this sentence, as the cause of brain A β accumulation in AD patients is still not clear. It could be a combination of lower clearance

or higher production. In familial cases, it seems to be due to the alteration of the carboxypeptidase activity of the gamma-secretase complex due to mutations in presenilin, that triggers the increased production of longer and more toxic Abeta species (therefore, not affecting APP-C99 levels) (Chavez-Gutierrez et al., EMBO J, 2012).

This sentence has been modified following the referee's advice.

3) In the introduction, it is stated that: "Further linking AD and cholesterol, APP-CTFs processing occurs in lipid rafts (14)" Authors need to correct this sentence, as processing can occur in other detergent-resistant microdomains (DRM), such as tetraspanin-enriched microdomains (Wakabayashi et al, Nat. Cell. Biol., 2009). A simple solution would be to replace the term "lipid raft" by DRMs.

We thank the reviewer for this suggestion. We have replaced the term *lipid raft* by DRMs.

4) In Figure 1E-F, it is shown that PS-DKO cells contain higher levels of fluorescently-labeled cholesterol, which is attributed to increased uptake. An alternative to increased uptake is reduced removal from endosomes/MVBs. In light of the argument that C99 increases uptake, authors need to demonstrate that cholesterol efflux from endosomes/MVBs remains unaffected.

We thank the reviewer for raising this interesting point. To address this issue, we performed a pulse-chase assay where we incubated the MEF cellular models that accumulate C99 with ³H-cholesterol. One hour after incubation, cells were washed to remove the excess of exogenous radioactive cholesterol and incubated in unlabeled media. After the indicated post-incubation times, levels of radioactivity in the media were measured for each of the conditions shown. Our data shown below indicates that cholesterol efflux correlates with increased cholesterol uptake, suggesting that, rather than a reduced removal, these phenotypes are caused by an increase in cholesterol uptake. The new data is included as Fig. S1I and S3E.

A) Cholesterol efflux, at 2, 4 and 6h, upon 1h pulse-chase with ³H-cholesterol was assayed in WT or PS-DKO cells previously treated for 16h with DAPT or a BACE inhibitor (BI), respectively. DMSO was used as a vehicle. **(B)** APP-DKO or WT (APP-WT) were transfected with an empty vector (EV) or the constructs C99^{WT} or C99^{MUT} and, 24h after transfection and DAPT incubation to prevent the cleavage of C99, cholesterol efflux was assayed as in (A).

5) In this sense, in Figures 2C-F it is shown that cholesterol uptake is not reversed by SMase inhibition in either DAPT-treated WT cells (Fig. 2C-D), or in PS-DKO cells (Fig. 2E-F). Ceramide triggers budding of exosome vesicles into multivesicular endosomes, which are enriched in cholesterol. Thus, inhibition of SMase could trigger the accumulation of cholesterol.

Exosome formation requires the activation of SMase activity. In fact, one of the sphingomyelinase inhibitors used in this report (GW4869) is regularly used to block the synthesis of MVB/exosomes in the ER. Therefore, it is quite unlikely that, in our experimental conditions, the enlarged filipin-positive droplets we observe in γ -secretase-deficient cells are the result of an accumulation of cholesterol in MVB/exosomes. We believe that incubation with SMase inhibitors phenocopies the alterations in cholesterol trafficking shown in cellular models of Niemann-Pick disease, where reductions in SMase activity impede the delivery of cholesterol to the ER. Under these conditions, the Insig-Scap-SREBP-2 pathway would be activated, and the uptake of extracellular cholesterol increased.

6) The order of appearance of some figures in the text is altered. The references to the figures in the text jump from Figure 2 to Figure 5: example: "While C99WT showed a perinuclear pattern of colocalization with ER and mitochondria, C99CBD presented a less marked perinuclear localization and a decreased association with mitochondria (Figs. 5A, 5B, and S5A)."

We apologize for this mistake. We have renumbered the figures according to the order of appearance.

7) The co-localization between C99wt and mitochondrial and ER markers is not obvious in Figures 3A and Figure 5A. Maybe it would help to show a zoom in Figure 5A.

We have modified the figures to include higher magnification insets that illustrate our conclusions.

Referee #3:

The manuscript by Pera et al. investigates how a fragment of APP control cholesterol homeostasis between the ER and the plasma membrane. The manuscript is situated in the control of lipid raft formation, a mechanism that depends on the enrichment of cholesterol under the control of sphingomyelin. The activities and amounts of cholesterol are tightly regulated at the level of the ER via interactions with the plasma membrane and mitochondria at the MAMs. In the manuscript, Pera et al. provide new data on the processing of APP, whose C99 fragment is generated on intercellular lipid rafts. The authors had previously shown that this processing localizes to MAMs, ER-associated lipid rafts in the proximity of mitochondria.

In this new manuscript, they now provide evidence that C99 is required for the lipid/cholesterol-detoxifying activities of MAMs as one of the functions of this ER domain.

The manuscript currently falls short of looking beyond these functions. However, given the known, well-characterized function of MAMs in multiple signaling events, the investigation of MAM functions should not be limited to lipid homeostasis. Instead, some aspect of other MAM functions should be added to gain a better understanding of what is going on.

Another area of deficiencies is a current lack of link with tethering mechanisms. At the moment, there are also some inconsistencies in data presentation, i.e., some assays are done for some conditions, but not others. Nevertheless, this is an important topic that warrants high profile exposure.

Major Points

1. Some MAM signaling functions beyond lipid homeostasis should be investigated to some extent. For instance, calcium transfer from the ER to mitochondria should be shown. This is critical, since we do not know very well how these diverse functions interact with each other.

2. Are tethers affected by C99 and its production? One way to address this could be via analysis of tether complex co-immunoprecipitation or via targeting of tethers to MAMs.

Specific Points:

1. In figure 1, there is inconsistency of data presentation between 1A and 1B. The % molarity is very different between the two. It is also unclear that 1B is total homogenate. Why was this not analyzed for the fractions like 1A?

As shown in the legend of Fig. 1, we have stated that total homogenates were used for this analysis. The purpose of this experiment was to compare the effect of different APP fragments on the overall cellular cholesterol content, and confirm our previous findings (Pera et al, 2017) on the role of C99 in MAM regulation and cholesterol metabolism. A subsequent analysis of the impact of C99 on the content of cholesterol in subcellular fractions from APP-DKO cells and controls is shown in figure 5E-H.

Mol% represents the moles of any given lipid normalized by all lipids extracted. This value does not reflect the absolute % of this lipid in the overall lipid composition of the cell, as ALL cellular lipid cannot be extracted for analysis. Therefore, it just represents a relative proportion of this lipid in the assayed cells and tissues. To avoid confusing the reader, we have decided to report these values as % of controls. As shown in the new Fig 1A and 1B, this does not affect the results.

2. A loading control is missing in Figure S1C.

We apologize for the confusion. This has been corrected. A loading control is included now.

3. What is the cholesterol uptake in APP DKO and transfectants of C99? Figure 1 just shows PS DKO. We need to understand the relative defects of the cellular models used. This is inconsistent presentation.

Cholesterol uptake was analyzed in control and transfected APP-DKO cells expressing WT or mutant forms of C99 (Fig. 5F). Our data shows that, contrary to those expressing C99^{MUT}, APP-DKO cells expressing C99^{WT} present with increases in the uptake (Fig. 5F) and esterification (Fig. 5G) of exogenous cholesterol.

4. A description of the filipin staining pattern via co-localization with cellular markers in the PS-DKO is necessary.

Following the referee's suggestion, we have analyzed whether the filipin puncta in PS-DKO cells or DAPT-treated WT cells colocalize with markers of different organelles in Supp. Fig. S1K. Our data shows that filipin puncta colocalize mainly with endosomal (Rab7) and lysosomal markers (LAMP1). As expected, some diffuse filipin staining can also be found in ER (Sec61 β) or Golgi (GM130).

5. Supplemental figure 3 should contain zoomed-in areas, similar to Figure 3.

We have modified the figures to include higher magnification insets that illustrate our conclusions.

6. The gradient fractionation should contain pan-ER and mitochondrial markers, to allow for judging where the analyzed proteins have fractioned to.

We have included pan-ER and mitochondrial markers in the western blot analysis of our sucrose gradient fractionation (Fig. 3). Calreticulin and ERp72 were not detected on the sucrose gradient fractions, confirming the absence of cross-contamination with ER markers.

7. It is currently impossible to understand where the APP DKO MAM formation stands relative to wild type MEFs. This control should be added to Figure 5B.

We apologize for this. We have included the values obtained from APP^{WT} cells as a control:

- Fig 1B (blue column). APP-WT levels were used as reference to represent the other groups
- In Fig. 4D (dashed line represents WT levels). APP-DKO treated with DAPT showed no changes in the levels of ER-mitochondria association when compared to APP-WT treated with DAPT.
- In Fig. 4E (APP WT are represented in blue columns). The data were referred to the levels of APP-WT at 2h for normalization. APP-DKO treated with DAPT showed a significant reduction in phospholipid transfer activity when compared to APP-WT treated with DAPT.
- In Fig. 5E, 5F and 5G (dashed line represents APP WT levels). The data were referring to the levels of APP-WT for normalization.
- In Fig. S3E. Cholesterol efflux data. APP-WT column is shown.
- In Fig. S3F (blue groups). SMase activity upon DAPT treatment was significantly increased in APP-WT while DAPT treatment showed no changes in APP-DKO (black columns)

Also, would the chemical removal of cholesterol show a similar effect in wild type cells (that would not be observed in DKO)?

This is an interesting point. Our data indicates that the absence of C99 in APP-DKO cells results in the reduction of cholesterol trafficking from PM to ER, thereby impeding the formation of MAM domains.

It is known that increases in cholesterol in PM induce the amyloidogenic APP pathway by favoring the cleavage of APP by BACE1 versus that of α -secretase. Thus, we would expect that chemical removal of cholesterol in WT cells will increase the localization of APP in PM and as well as its cleavage via the non-amyloidogenic pathway. Interestingly, it has been recently shown (DeBove *et al*, 2019) that chemical removal of cholesterol by methyl- β -cyclodextrin indeed favors α -secretase cleavage in WT cells, but without affecting β/γ -secretase activity.

In light of these data and our experience, we believe that the chemical removal of cholesterol in WT cells first impacts on the composition of the plasma membrane, before reaching intracellular membranes. However, the acute decreases in cholesterol levels in PM will rapidly elicit feedback

mechanisms to replenish the membranes with newly synthesized or internalized cholesterol. In light of our data, we believe that the chemical removal of cholesterol in APP-DKO cells might result in an increased in the *de novo* synthesis of cholesterol, but it would not rescue the defects in cholesterol trafficking between PM-ER, nor in MAM functionality. This is the focus of our current studies. Similarly, it is possible that the chemical removal of cholesterol in PS-DKO cells increases the cleavage by α -secretase. However, as shown (DelBove *et al.*, 2019), it will neither alleviate the upregulation of the β/γ -secretase activity, nor the activation of MAM turnover.

Minor concerns:

1. The first paragraph of the Results section is difficult to read.

This paragraph has been edited for clarity.

2. What does the abbreviation CTFs stand for?

C-terminal fragments. This abbreviation has been included in the text.

References

- Das A, Brown MS, Anderson DD, Goldstein JL, Radhakrishnan A (2014) Three pools of plasma membrane cholesterol and their relation to cholesterol homeostasis. *Elife* 3
- DelBove CE, Strothman CE, Lazarenko RM, Huang H, Sanders CR, Zhang Q (2019) Reciprocal modulation between amyloid precursor protein and synaptic membrane cholesterol revealed by live cell imaging. *Neurobiol Dis* 127: 449-461
- Endapally S, Frias D, Grzemska M, Gay A, Tomchick DR, Radhakrishnan A (2019) Molecular Discrimination between Two Conformations of Sphingomyelin in Plasma Membranes. *Cell* 176: 1040-1053 e1017
- Guo Q, Fu W, Sopher BL, Miller MW, Ware CB, Martin GM, Mattson MP (1999) Increased vulnerability of hippocampal neurons to excitotoxic necrosis in presenilin-1 mutant knock-in mice. *Nat Med* 5: 101-106
- Pera M, Larrea D, Guardia-Laguarta C, Montesinos J, Velasco KR, Agrawal RR, Xu Y, Chan RB, Di Paolo G, Mehler MF *et al* (2017) Increased localization of APP-C99 in mitochondria-associated ER membranes causes mitochondrial dysfunction in Alzheimer disease. *EMBO J* 36: 3356-3371
- Pierrot N, Tyteca D, D'Auria L, Dewachter I, Gailly P, Hendrickx A, Tasiaux B, Haylani LE, Muls N, N'Kuli F *et al* (2013) Amyloid precursor protein controls cholesterol turnover needed for neuronal activity. *EMBO Mol Med* 5: 608-625
- Slotte JP, Bierman EL (1987) Movement of plasma-membrane sterols to the endoplasmic reticulum in cultured cells. *Biochem J* 248: 237-242
- Slotte JP, Bierman EL (1988) Depletion of plasma-membrane sphingomyelin rapidly alters the distribution of cholesterol between plasma membranes and intracellular cholesterol pools in cultured fibroblasts. *Biochem J* 250: 653-658

Dear Estela,

Thank you for submitting your revised manuscript to The EMBO Journal. Your study has now been seen by the three referees and their comments are provided below.

As you can see from the comments, the referees appreciate the introduced changes and support publication here. I am therefore very pleased to let you know that we can accept the manuscript for publication here.

Before sending you the formal acceptance letter there are just a few issues to be resolved in a final revision:

- You can only have 5 keywords
- We also need a data availability section. As far as I can see you don't have any data that needs to be deposited in an external data base if correct then please add Data Availability: This study includes no data deposited in external repositories
- Please re-label "Author Information" as "Conflict of interest"
- We don't allow data not shown (pg 15) - can you maybe add the data to the appendix?
- We also need a checklist - please see guide to authors.
- Corresponding author(s) need to register for an ORCID ID and link it to the journal account.
- The figure files need to be uploaded as separate figure files
- The supplemental files should either be labelled as Expanded View figures (if so then they need to be uploaded as individual files) or added to an appendix (one file with a ToC) - see also guide to authors. Please correct callouts in the text to the supplemental figures
- There are 2 tables in the Article file called Table A and Table 6, They should be re-labelled Tables 1 and 2 please correct callout in text.
- We need a word file of the manuscript text as well. Can you make sure the order of the sections is OK?
- We include a synopsis of the paper (see <http://emboj.embopress.org/>). Please provide me with a general summary statement and 3-5 bullet points that capture the key findings of the paper.
- We also need a summary figure for the synopsis. The size should be 550 wide by [200-400] high (pixels). You can also use something from the figures if that is easier.

That should be all - let me know if you have any further questions.

with best wishes

Karin

Karin Dumstrei, PhD
Senior Editor
The EMBO Journal

Further information is available in our Guide For Authors:

The revision must be submitted online within 90 days; please click on the link below to submit the revision online before 1st Oct 2020.

Referee #1:

This revised manuscript took into account all our comments. It now presents solid evidence for the conclusion that chol trafficking between ER and mitochondria can be perturbed by mutated C99.

The implications for AD are important and therefore the paper is a strong candidate to be published in the EMBOJ.

Referee #2:

The authors have answered all questions satisfactorily. In the new version of the manuscript, they include experiments in cultured cortical neurons from WT and PS1M146V-KI mice, an AD mouse model, which support the previous fibroblasts (MEF) data. They also include a pulse-chase experiment that helps to exclude an alteration of cholesterol efflux. Finally, they have amended the mistakes in the text such as the altered numbering of figures. All in all, the manuscript has improved substantially,

Referee #3:

The manuscript by Pera et al. has been significantly improved. All of my major points with the exception of one have been taken care of. At this point, the manuscript provides no information about calcium transfer from the ER to mitochondria, nor information whether calcium-regulatory proteins have been affected. This is a regrettable lack of information that could have been addressed (e.g., localization of calcium-regulatory proteins on the sucrose gradient). Unless other reviewers agree this should be looked at, I am ok with the authors addressing this deficiency in the discussion.

Dear Estela,

Thanks for submitting your revised manuscript to The EMBO Journal. I have now had a chance to take a look at everything and all looks good.

I am therefore very happy to accept the manuscript for publication here.

Congratulations on a nice study!

with best wishes Karin

Karin Dumstrei, PhD
Senior Editor
The EMBO Journal

Please note that it is EMBO Journal policy for the transcript of the editorial process (containing referee reports and your response letter) to be published as an online supplement to each paper. If you do NOT want this, you will need to inform the Editorial Office via email immediately. More information is available here: http://emboj.embopress.org/about#Transparent_Process

Your manuscript will be processed for publication in the journal by EMBO Press. Manuscripts in the PDF and electronic editions of The EMBO Journal will be copy edited, and you will be provided with page proofs prior to publication. Please note that supplementary information is not included in the proofs.

Should you be planning a Press Release on your article, please get in contact with embojournal@wiley.com as early as possible, in order to coordinate publication and release dates.

If you have any questions, please do not hesitate to call or email the Editorial Office. Thank you for your contribution to The EMBO Journal.

** Click here to be directed to your login page: <http://emboj.msubmit.net>

Corresponding Author Name: Estela Area-Gomez

Manuscript Number: 103791